# Size Transferability of Graph Transformers with Convolutional Positional Encodings

## Abstract

Transformers have achieved remarkable success across domains, motivating the rise of Graph Transformers (GTs) as attention-based architectures for graph-structured data. A key design choice in GTs is the use of Graph Neural Network (GNN)-based positional encodings to incorporate structural information. In this work, we study GTs through the lens of manifold limit models for graph sequences and establish a theoretical connection between GTs with GNN positional encodings and Manifold Neural Networks (MNNs). Building on transferability results for GNNs under manifold convergence, we show that GTs inherit transferability guarantees from their positional encodings. In particular, GTs trained on small graphs provably generalize to larger graphs under mild assumptions. We complement our theory with extensive experiments on standard graph benchmarks, demonstrating that GTs exhibit scalable behavior on par with GNNs. To further show the efficiency in a real-world scenario, we implement GTs for shortest path distance estimation over terrains to better illustrate the efficiency of the transferable GTs. Our results provide new insights into the understanding of GTs and suggest practical directions for efficient training of GTs in large-scale settings.

## 1. Introduction

Transformers have recently been adapted to graph-structured data by injecting graph information through positional or structural encodings while retaining global self-attention—yielding Graph Transformers. Graph transformers have delivered state-of-the-art or highly competitive results in several domains, including but not limited to large-scale molecular property prediction (Ying et al., 2021), biomedical knowledge graphs (Hu et al., 2020), and long-range data benchmark (Dwivedi et al., 2022).

Graph transformers perform attention on sets of nodes not strictly limited by their local neighborhoods, and incorporate structural information via positional encodings. Early examples compute dense, pairwise attention over all nodes of the graph, such as (Dwivedi & Bresson, 2020) which uses the top$-k$ eigenvectors of the graph Laplacian as positional encodings, and SAN, which learns spectral encodings drawn from the Laplacian spectrum (Kreuzer et al., 2021). Other graph transformers use relative (pairwise) positional encodings, such as GraphiT (diffusion-kernels) (Mialon et al., 2021), the Graphormer family (shortest-path and centrality biases in dense self-attention) (Ying et al., 2021), random walk approaches (Ma et al., 2023; Geisler et al., 2023), or GKAT (heat kernel-based) (Choromanski et al., 2022). Mask-based relative encodings replace quadratic attention with structured sparsity to trade off globality and scalability, for example Exphormer (local neighborhood + expander graphs) (Shirzad et al., 2023), and UnifiedGT ($k$-hop neighborhood) (Lin et al., 2024). We refer to (Black et al., 2024) for an in-depth analysis on the properties of absolute and relative positional encodings on graph transformers.

A central insight across this literature is that structure must be encoded explicitly for attention to be effective on graphs. The outstanding performance of full pairwise attention is restricted by its inherent computational complexity, impeding scalability to graphs beyond hundreds of thousands of nodes. Moreover, data collection costs for training with large graphs can be prohibitively expensive. Therefore, it is paramount to develop a graph transformer that can be *transferable* across different graph sizes under theoretical guarantees.

Theory from spectral graph signal processing establishes that graph convolutional filters—under continuity conditions—are stable to perturbations and transferable across graphs sampled from the same limit model (Wang et al., 2024b;c; Ruiz et al., 2023; 2020; Keriven et al., 2020). Fur-

[1]Anonymous Institution, Anonymous City, Anonymous Region, Anonymous Country. Correspondence to: Anonymous Author <anon.email@domain.com>.

Preliminary work. Under review by the International Conference on Machine Learning (ICML). Do not distribute.

*Figure 1.* Diagram of Graph Transformer (GT) with RPEARL Positional Encodings. The graph $\mathbf{G}$ is sampled from manifold $\mathcal{M}$. The graph structure is processed by RPEARL using a graph neural network. The positional encodings are added to the node features and passed to the graph transformer, which outputs node features $\mathbf{Y}$.

thermore, by analyzing graphs through limits, one obtains that graph filters and GNNs converge as graph size grows; hence, models trained on small graphs can be deployed on larger graphs from the same limit model without retraining.

Building on this theory (Kanatsoulis et al., 2024) has proposed RPEARL, which uses GNNs with random features as positional encodings. They show that an architecture consisting of RPEARL and a GNN backbone is provably more expressive than a message-passing GNN. RPEARL encodings are permutation equivariant and retain the same stability and transferability of GNNs. We show that when these stable and transferable encodings are fed to a transformer whose attention is controlled to be Lipschitz—e.g., by normalization schemes for self-attention or by alternative Lipschitz attention maps—the composed model inherits stability and size-transferability. Practically, this yields an efficient recipe: train on small graphs using graph convolutional positional encodings, then transfer to larger graphs while keeping attention regularized, achieving sub-linear performance difference and substantial computational savings.

The main contributions are as follows:

- We argue that GNNs are a principled choice of positional encodings to ensure the stability, equivariance, generalization, and transferability of the transformer inputs.

- We provide theoretical guarantees that graph transformers inherit the transferability properties of their GNN positional encodings, enabling competitive performance across different scales of graphs sampled from an underlying manifold without retraining.

- We propose a transferable, practical sparse graph

transformer with attention masking and RPEARL PEs, which is also covered by our transferability results.

- We carry out experiments where we verify the transferability of various graph transformer architectures on different domains (ArXiv-year, Reddit, snap-patents, MAG). We show that the performance and transferability of sparse GT can match or outperform GNNs and dense graph transformers. To the best of our knowledge, our experiments provide the first results where full batch performance of graph transformers (dense and sparse) is analyzed on graphs larger than 1M nodes.

- We validate our Sparse Graph Transformer over real-world terrain datasets and solve the shortest path distance estimation problem.

### 1.1. Related work

**Transferability and Generalization via Limit Models.**
The convergence of GNNs to manifold neural networks (MNNs) has been leveraged extensively as an analytical tool to establish the transferability, stability, and statistical generalization of graph neural networks (Wang et al., 2024b;c;a). Transferability has also been explored with graphons as a limit model for graphs (Ruiz et al., 2020; Maskey et al., 2025; 2023). More recently, (Levin et al., 2025) studied a general framework for transferability by relating small graph instances to limit objects. (Li et al., 2023) presented the generalization analysis for general transformers without considering graph structural information.

**Efficent training on large graphs.** The seminal work on efficient graph training is GraphSage (Hamilton et al.,

2017) scales GNNs to large-scale graphs by training on minibatches of neighborhood subgraphs. (Lin et al., 2024) leverages GraphSage-style sampling. Based on graphon convergence results, (Cerviño et al., 2023) proposes training on growing graphs.

**Length Generalization in Transformers.** Length generalization in large language models has been studied both empirically (Zhou et al., 2024; Anil et al., 2022) and theoretically (Huang et al., 2024). It has been argued that positional encodings play an important role in achieving robust length generalization (Kazemnejad et al., 2023; Peng et al., 2023) Yet, the characterization and necessary conditions for length generalization remain an open area of research.

**GNN-Transformer Hybrids.** Several works have proposed similar hybrid "local-global" models by combining message-passing GNNs with global attention. For instance, GraphGPS (Rampášek et al., 2022), SGFormer (Wu et al., 2023), GraphTrans (Wu et al., 2022), and UnifiedGT (Lin et al., 2024). The theoretical results on the transferability of graph transformers also covers these hybrid architectures.

# 2. Graph Transformer With RPEARL Positional Encodings

**Set up.** An undirected graph $\mathbf{G} = (\mathcal{V}, \mathcal{E}, \mathcal{W})$ contains a node set $\mathcal{V}$ with $N$ nodes and an edge set $\mathcal{E} \subseteq \mathcal{V} \times \mathcal{V}$. The weight function $\mathcal{W} : \mathcal{E} \to \mathbb{R}$ assigns values to the edges. We define the graph Laplacian $\mathbf{L} = \text{diag}(\mathbf{A1}) - \mathbf{A}$ where $\mathbf{A} \in \mathbb{R}^{N \times N}$ is the weighted adjacency matrix. Graph signals are functions mapping nodes to feature vectors, which we denote by $\mathbf{x}_i \in \mathbb{R}^D$ and collect in a matrix of column vectors $\mathbf{X} = [\mathbf{x}_1, \ldots, \mathbf{x}_N]$.

## 2.1. Graph Transformer

A graph transformer is a layered architecture that processes graph signals $\mathbf{X}$ and outputs a graph signal matrix $\mathbf{Y} \in \mathbb{R}^{D \times N}$ by using an attention (Vaswani et al., 2017) operation, where each layer computes

$$\mathbf{Y} = \mathbf{\Phi_G}(\mathcal{T}, \mathbf{X}) = \mathbf{VX} \, \text{softmax} \left[ (\mathbf{QX})^\top (\mathbf{KX}) \right]. \quad (1)$$

The graph transformer $\mathbf{\Phi_G}$ is parameterized by the query, key and value linear maps $\mathbf{Q}, \mathbf{K}, \mathbf{V} \in \mathbb{R}^{D \times D}$, collected in $\mathcal{T} = \{\mathbf{Q}, \mathbf{K}, \mathbf{V}\}$. The operation $(\mathbf{QX})^\top (\mathbf{KX})$ computes inner products between pairs of projected node features $\langle \mathbf{Qx}_i, \mathbf{Kx}_j \rangle$, for all $i, j \in [1, N]$. Then, the softmax operation normalizes the pairwise inner products so that the rows sum to one. Finally, the output node features are computed as a linear combination of the projected values weighted by the attention coefficients.[1] The

---

[1] For simplicity of presentation, we omit the transformer's feedforward layer, softmax normalization constant, and low-rank

input to the graph transformer is $\mathbf{X} + \mathbf{\Psi_G}(\mathcal{H}, \mathbf{L}, \mathbf{Z})$, where $\mathbf{\Psi_G} : \mathbb{R}^{N \times D} \to \mathbb{R}^{N \times D}$ is called the graph *positional encoder*. The operator $\mathbf{\Psi_G}$ maps each node into a positional encoding vector utilizing the structural information of $\mathbf{L}$. Therefore, an effective choice of $\mathbf{\Psi_G}$ must satisfy key properties for graph signal processing, such as permutation equivariance, expressivity, and transferability.

## 2.2. Positional encodings with Graph Neural Networks

Consider using a graph neural network (GNN) as a learnable positional encoder $\mathbf{\Psi_G}$. A GNN is a layered architecture composed of graph convolutional filters followed by pointwise nonlinearities. One layer of a GNN is written as

$$\mathbf{P} = \sigma \left[ \sum_{k=0}^{K-1} \mathbf{H}_k \mathbf{ZL}^k \right] \quad (2)$$

where $\mathbf{Z} \in \mathbb{R}^{D \times N}$ are the input signals, and $\mathcal{H} = \{\mathbf{H}_1, \ldots, \mathbf{H}_K\}$ collects the filter coefficients. Equation (2) performs a graph convolution of order $K$ on the graph signals by the graph Laplacian $\mathbf{L}$, followed by the pointwise nonlinearity $\sigma$. The input signal $\mathbf{Z}$ can be the graph signals, i.e. $\mathbf{Z} = \mathbf{X}$. In the absence of data, it is also possible to leverage random features. Draw $m \in [1, M]$ realizations $\mathbf{z}^{(m)}$ of an $N$-dimensional random variable from an isotropic Gaussian distribution $\mathbf{z} \sim \mathcal{N}(\mathbf{0}, \mathbf{I}_N)$, and pass each realization in parallel to the GNN:

$$\mathbf{P}^{(m)} = \sigma \left[ \sum_{k=0}^{K-1} \mathbf{H}_k \mathbf{z}^{(m)\top} \mathbf{L}^k \right], \quad (3)$$

where each $\mathbf{H}_k \in \mathbb{R}^{D \times 1}$. This results in $M$ independent output matrices $\mathbf{P}^{(m)} \in \mathbb{R}^{N \times D}$. The realizations $\mathbf{z}^{(m)}$ play the role of random node IDs, which are known to improve the expressiveness of the GNN by breaking structural symmetries (You et al., 2021). However, the resulting outputs $\mathbf{P}^{(m)} \in \mathbb{R}^{N \times D}$ lose permutation equivariace. Noting that the distribution of $\mathbf{z}$ and its statistics are themselves permutation equivariant, we can recover this property by taking their empirical average:

$$\mathbf{\Psi_G}(\mathcal{H}, \mathbf{L}, \mathbf{Z}) = \hat{\mathbb{E}} \left[ \mathbf{P}^{(1)}, \ldots, \mathbf{P}^{(M)} \right] = \frac{1}{M} \sum_{m=1}^{M} \mathbf{P}^{(m)}. \quad (4)$$

The output is the positional encoding $\mathbf{\Psi_G}(\mathcal{H}, \mathbf{L}, \mathbf{Z})$ with dimensions $N \times D$. This positional encoder, named RPEARL, is shown by (Kanatsoulis et al., 2024) to provably enhance the expressive power of GNNs beyond the WL-test while preserving key GNN properties (stability, scalability, and transferability) (Ruiz et al., 2021).

---

projections. While we present our theory on a single-layer, single-head transformer, its extension to these settings graph transformers is straightforward.

# 3. Transferability Analysis of Graph Transformers via a Manifold Perspective

A central bottleneck of graph transformers lies in their efficient deployment on large-scale graphs. The quadratic complexity of attention relative to graph size makes naive scaling impractical, often requiring costly retraining. Transferability offers a principled way to overcome this limitation by enabling models to capture structural regularities that persist across graph sizes and instances. In particular, if a sequence of graphs admits a well-defined limit object, then a model trained on small-scale graphs can, in principle, be transferred to larger-scale graphs without retraining, while preserving its functional behavior. In this section, we show that the transferability of graph transformers is inherited from the transferability of their positional encodings. That is, once positional encodings are size-transferable functions over the graph limit, the induced graph transformer naturally generalizes across graph scales. Building on this insight, we present the first transferability analysis of graph transformers from a manifold-based perspective, establishing a theoretical foundation for size-generalizable and scalable graph transformers.

## 3.1. Discrete graphs and operator limits

We study transferability by relating graph-based computations to their continuum counterparts on a manifold. Let $\mathcal{M}$ be a $d$-dimensional compact, smooth and differentiable Riemannian submanifold embedded in a M-dimensional space $\mathbb{R}^M$ with finite volume. This induces a measure $\mu$ which has a non-vanishing Lipschitz continuous density $\rho$ with respect to the Riemannian volume over the manifold with $\rho : \mathcal{M} \to (0, \infty)$, assumed to be bounded as $0 < \rho_{min} \leq \rho(x) \leq \rho_{max} < \infty$ for all $x \in \mathcal{M}$. The manifold data supported on each point $x \in \mathcal{M}$ is defined by vector-valued functions $f : \mathcal{M} \to \mathbb{R}^p$ (Wang et al., 2024c). We use $L^2(\mu; \mathbb{R}^p)$ to denote square-integrable $\mathbb{R}^p$-valued functions over $\mathcal{M}$ with respect to measure $\mu$.

Given a set of $N$ i.i.d. samples $X_N = \{x_i\}_{i=1}^N$ drawn from $\mu$ over $\mathcal{M}$, we construct a graph $\mathbf{G}(\mathcal{V}, \mathcal{E}, \mathcal{W})$ on these $N$ sampled points $X_N$, where each point $x_i$ is a vertex of graph $\mathbf{G}$, i.e. $\mathcal{V} = X_N$. Each pair of vertices $(x_i, x_j)$ is connected with an edge while the weight attached to the edge $\mathcal{W}(x_i, x_j)$ is determined by a kernel function $K_\epsilon$. The kernel function is decided by the Euclidean distance $\|x_i - x_j\|$ between these two points. The graph Laplacian denoted as $\mathbf{L}$ can be calculated based on the weight function (Merris, 1995). The constructed graph Laplacian with an appropriate kernel function has been proved to approximate the Laplace operator $\mathcal{L}$ of $\mathcal{M}$ (Calder & Trillos, 2022; Belkin & Niyogi, 2008; Dunson et al., 2021). In this paper, we implement the normalized Gaussian kernel definition in (Dunson et al., 2021).

To compare discrete and continuous computations over the sampled graphs and ove the manifold, we introduce two linear mappings that connect data from sampled graph $\mathbf{G}_N$ and manifold $\mathcal{M}$ and back, i.e. sampling operator as $\mathbf{S}_N : L^2(\mu; \mathbb{R}^p) \to \mathbb{R}^{N \times p}$ and interpolation operator as $\mathbf{I}_N : \mathbb{R}^{N \times p} \to L^2(\mu; \mathbb{R}^p)$. We assume that these operators are bounded and consistent (Wang et al., 2024b; Levie et al., 2021), i.e.

$$\lim_{N \to \infty} \|\mathbf{I}_N \mathbf{S}_N f - f\|_{L^2(\mu)} = 0, \text{ for all } f \in L^2(\mu; \mathbb{R}^p).$$
(5)

With these mapping operators, the convergence of graph Laplacians $\mathbf{L}_N$ over constructed graphs $\mathbf{G}_N$ to the continuous operator $\mathcal{L}$ can be written as

$$\lim_{N \to \infty} \|\mathbf{I}_N \mathbf{L}_N \mathbf{S}_N f - \mathcal{L}f\|_{L^2(\mu)} = 0,$$
(6)

for all $f \in L^2(\mu; \mathbb{R}^p)$.

## 3.2. Transferable Functions over Manifolds

We now formalize transferability as a property of functions defined through graph operators that remain well defined in the continuum limit.

Let $\mathbf{\Psi}$ be a map defined on any Hilbert space with a bounded linear operator. We write this dependence explicitly as $\mathbf{\Psi}_\mathbf{G}(\mathbf{L}_N, \mathbf{X}) \in \mathbb{R}^{N \times q}$ with $\mathbf{X} \in \mathbb{R}^{N \times p}$ as a graph-level realization and $\mathbf{\Psi}_\mathcal{M}(\mathcal{L}, f) \in L^2(\mu; \mathbb{R}^q)$ with $f \in L^2(\mu; \mathbb{R}^p)$ as a manifold-level realization. The same functional form $\mathbf{\Psi}$ is thus instantiated on discrete graphs and on the continuum, with dependence on the underlying domain entering solely through the operator $\mathbf{L}_N$ or $\mathcal{L}$.

**Definition 1** (Transferable functions over Manifolds). *Assume that $\mathbf{\Psi}$ is permutation equivariant and uniformly Lipschitz in its inputs. $\mathbf{\Psi}$ is said to be size-generalizable with respect to the operator convergence $\mathbf{L}_N \to \mathcal{L}$ if for every admissible input function $f \in L^2(\mu; \mathbb{R}^p)$,*

$$\lim_{N \to \infty} \left\| \mathbf{I}_N \mathbf{\Psi}_\mathbf{G}(\mathbf{L}_N, \mathbf{S}_N f) - \mathbf{\Psi}_\mathcal{M}(\mathcal{L}, f) \right\|_{L^2(\mu)} = 0. \quad (7)$$

Definition 1 formalizes the notion of transferability as a function whose output converges to the limit as operating on the manifold. As a consequence, a transferable function $\mathbf{\Psi}$ can be trained or defined on finite graphs of one size and deployed on other graph instances—possibly of different sizes or resolutions—as long as the associated operators converge to the same manifold limit $\mathcal{L}$.

This operator-centric perspective underlies our analysis of graph transformers in the sequel, where we show that their transferability is inherited from the transferability of the positional encodings that define their operator dependence.

### 3.3. Transferable Graph Transformers

To study the properties of graph transformers, we first define the counterpart of transformers over the manifold as the limit object.

**Definition 2** (Manifold Transformer (MT))**.** *A manifold transformer layer is defined as*

$$\mathbf{\Phi}_{\mathcal{M}}(\mathbf{T}, f)(x) = \frac{\int_{\mathcal{M}} e^{\langle \mathbf{Q}f(x), \mathbf{K}f(y)\rangle} \mathbf{V}f(y) d\mu(y)}{\int_{\mathcal{M}} e^{\langle \mathbf{Q}f(x), \mathbf{K}f(y)\rangle} d\mu(y)}, \quad (8)$$

*for all $x \in \mathcal{M}$ with the input function $f$ defined as a positional encoding of a manifold signal $g \in L^2(\mathcal{M})$*

$$f(x) = \mathbf{\Psi}_{\mathcal{M}}(\mathcal{H}, \mathcal{L}, g)(x). \quad (9)$$

Here, $f$ and $g$ are vector-valued functions over $\mathcal{M}$. The manifold transformer is a map $\mathbf{\Phi}_{\mathcal{M}} : L^2(\mu; \mathbb{R}^q) \rightarrow L^2(\mu; \mathbb{R}^D)$ resulting of the composition of $f$ (the positional encoding) with the manifold attention operation. Equation (9) corresponds to the evaluation of the limit object of the positional encodings over the input manifold signal $g$. Equation (8) specifies manifold attention, an integral over a weighted sum of the projected inputs, with each weight corresponding to the exponential of inner products of the manifold attention.

Equation (8) can be interpreted as a continuous analogue of the softmax attention in Equation (1) where attention coefficients are computed across infinitely many points across the manifold. We now introduce a set of assumptions required to ensure the convergence of GTs to MTs.

**Assumption 1** (Normalized Lipschitz signals)**.** *The input manifold signals $g$ are normalized Lipschitz for all points $a, b \in \mathcal{M}$, $\|g(b) - g(a)\| \le \|b - a\|$.*

**Assumption 2** (Bounded linear operators)**.** $\mathbf{Q}$, $\mathbf{K}$, *and* $\mathbf{V}$ *are bounded linear operators with constants* $C_Q, C_K, C_V > 0$, *i.e.,* $\|\mathbf{Q}x\| \le C_Q \|x\|$, $\|\mathbf{K}x\| \le C_K \|x\|$, $\|\mathbf{V}x\| \le C_V \|x\|$, *for all* $x \in \mathbb{R}^D$.

**Assumption 3** (Transferable positional encoding)**.** *The positional encoding function $\mathbf{\Psi}_{\mathbf{G}}(\mathcal{H}, \mathbf{X})$ is transferable and converges to a limit object $\mathbf{\Psi}_{\mathcal{M}}(\mathcal{H}, f) : \mathcal{M} \rightarrow \mathbb{R}^q$. The output difference between a transferable positional encoding and its limit object, for a point $x \in \mathcal{M}$, is denoted by $\|\mathbf{\Psi}_{\mathbf{G}}(\cdot)(x) - \mathbf{\Psi}_{\mathcal{M}}(\cdot)(x)\| \le \Delta_{\text{PE}}$*

Assumption 1 is a mild assumption on the underlying manifold. Assumption 2 can be seen as regularization of transformer operators. These can be realized in practice via penalty terms during training process. Assumption 3 is readily satisfied in the case where $\mathbf{\Psi}_{\mathbf{G}}$ is a GNN, which follows from established convergence results of GNNs to MNNs, shown via manifold convergence analysis in previous works (Wang et al., 2024b). Building on this foundation, in Appendix 6.2 we present a convergence theorem from the literature, tailored to our setting.

We can now present our main theorem on the convergence of GT to MT, which we will use to show the transferability of GT.

**Theorem 1.** *(Point-wise Convergence of GT to MT) For any $x \in \mathcal{M}$, under assumptions $1 - 3$, the pointwise output difference between a graph transformer and manifold transformer, with probability at least $1 - \delta$, is bounded by*

$$\|\mathbf{\Phi}_{\mathbf{G}}(\mathbf{T}, \mathbf{X})(x) - \mathbf{\Phi}_{\mathcal{M}}(\mathbf{T}, f)(x)\|_2 \le \Delta_{\mathbf{\Phi}(\mathbf{X})}$$
$$= [C_V + 4e^{2M}C_{QKV}]\Delta_{\text{PE}}$$
$$+ [e^M C_V + 2e^{2M}C_{QKV}]A\left(\frac{\log N}{N}\right)^{1/d}, \quad (10)$$

*where $A$ is a constant related to the geometry of $\mathcal{M}$, $d \ge 3$ is the intrinsic dimension of the manifold, $C_{QKV} = C_Q C_K C_V$ and $C_V$ are the linear operator bound constants of $\mathbf{Q}$, $\mathbf{K}$, and $\mathbf{V}$, and $M := \sup_{u,v \in \mathcal{M}} \langle \mathbf{Q}f(u), \mathbf{K}f(v)\rangle$.*

The proof of Theorem 1 is available in Appendix 6.6. This result proves that as the number of nodes sampled in graph $\mathbf{G}$ increases, the output of GT tends to converge to the underlying MT with a rate $\mathcal{O}\left((\log N/N)^{(1/d)}\right)$. The dependence of this result on the linear operator bounds $C_{CKV}$ and $C_V$, as well as the bound on the positional encoding $\Delta_{PE}$, suggests that smoother linear operators $\mathbf{Q}, \mathbf{K}, \mathbf{V}, \mathbf{H}$ can improve the convergence rate. This provides the insight that regularized operators in GT and its positional encoding helps to achieve a better convergence result, hence a better transferability performance.

Theorem 1 indicates that the pointwise output difference of GT and MT decays as the size of the sampled graph $N$ grows. This implies that we can train a GT on a small graph $\mathbf{G}_1$ with $N_1$ nodes, and *transfer* it to a larger graph $\mathbf{G}_2$, with $N_2 > N_1$, with guarantees that the approximation gap to the manifold transformer's output is bounded. This implication is meaningful in practice given the $\mathcal{O}(N^2)$ computational cost of GT – we can train on a relatively smaller graph and ensure good performance on larger graphs. We state this below in the following corollary:

**Corollary 2.** *(Transferability of Graph Transformers) Let graphs $\mathbf{G}_1$ and $\mathbf{G}_2$ constructed by points sampled from $\mathcal{M}$, with $N_1$, $N_2$ nodes respectively, and graph signals $\mathbf{X}_1$ and $\mathbf{X}_2$. Then, it holds, with probability $1 - \delta$,*

$$\frac{1}{\mu(\mathcal{M})}\|\mathbf{I}_{N_1}\mathbf{\Phi}_{\mathbf{G}_1}(\mathbf{T}, \mathbf{X}_1) - \mathbf{I}_{N_2}\mathbf{\Phi}_{\mathbf{G}_2}(\mathbf{T}, \mathbf{X}_2)\|_{L^{1,2}(\mathcal{M})}$$
$$\le \Delta_{\mathbf{\Phi}(\mathbf{X}_1)} + \Delta_{\mathbf{\Phi}(\mathbf{X}_2)}$$
$$+ 4e^{2M}C_{QKV}A\left(\frac{\log N}{N}\right)^{1/d} \quad (11)$$

Further, as we propose the architecture in Section 2, RPEARL as positional encodings also possess the transferability property as studied in (Kanatsoulis et al., 2024).

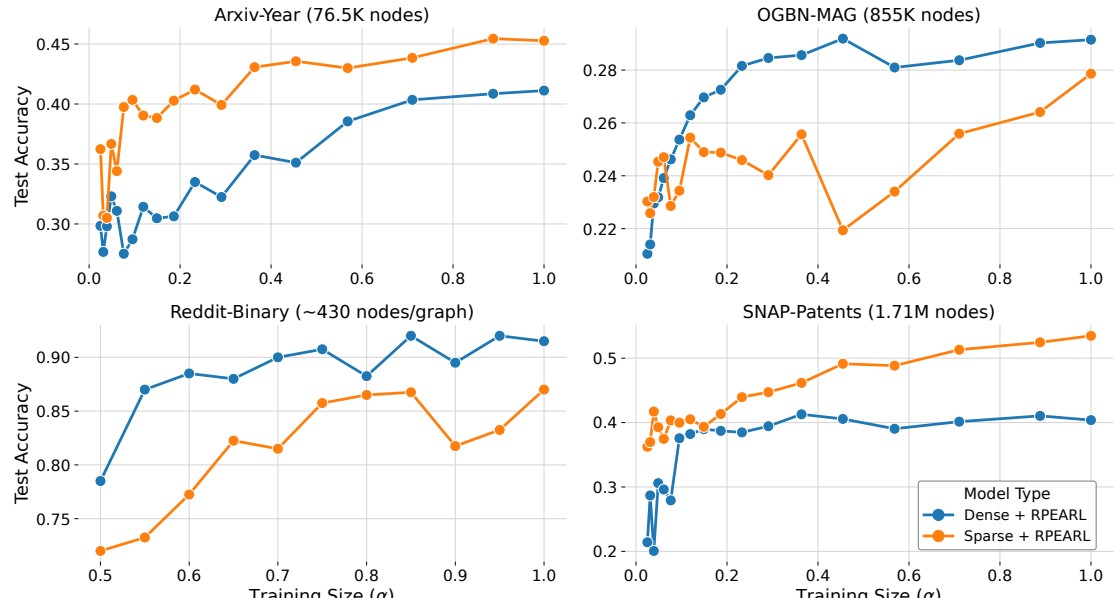

*Figure 2.* Transferability plots. For each dataset, the $x$ axis represents the train graph sizes as a proportion of the largest graph ($\alpha$), and the $y$ axis is the test accuracy at the full-sized graph. The titles show dataset name and largest graph size.

We state the transferability of graph transformers proposed in Section 2 as

**Corollary 3.** *Our proposed graph transformer architecture with RPEARL positional encoding is size tranferable with probability $1 - \delta$*

$$\|\mathbf{\Phi_G}(\mathbf{T}, \mathbf{X})(x) - \mathbf{\Phi}_{\mathcal{M}}(\mathbf{T}, f)(x)\|_2$$

$$\leq [C_V + 4e^{2M} C_{QKV}] \left( C_{\mathcal{M}} \left( \frac{\log(C/\delta)}{N} \right)^{\frac{1}{d+4}} \right.$$

$$+ \sqrt{\frac{\log(1/\delta)}{N}} \right) + [e^M C_V + 2e^{2M} C_{QKV}] A \left( \frac{\log N}{N} \right)^{1/d} \cdot \tag{12}$$

### 3.4. Sparse Graph Transformer (Sparse GT).

One common computational tradeoff in graph transformers is to mask coefficients to attend to a neighborhood of nodes reachable in $k$-hops instead of computing all pairwise node attentions. Restricting the attention operation to the $k$-hop neighborhood results in the Sparse Graph Transformer (Sparse GT) given by

$$\mathbf{x}_i = \frac{\sum_{j \in \mathcal{N}^{\leq k}(i)} \exp\{\langle \mathbf{Q}\mathbf{x}_i, \mathbf{K}\mathbf{x}_j \rangle\} \mathbf{V}\mathbf{x}_j}{\sum_{j \in \mathcal{N}^{\leq k}(i)} \exp\{\langle \mathbf{Q}\mathbf{x}_i, \mathbf{K}\mathbf{x}_j \rangle\}}. \tag{13}$$

where we denote the $k$-hop neighborhood of node $i$ as $\mathcal{N}^{\leq k}(i) = \bigcup_{k'=1}^{k} \mathcal{N}^k(i)$, with $\mathcal{N}^k(i) = \{j : j' \in \mathcal{N}(j), j' \in \mathcal{N}^{k-1}(i)\}$. We denote the Sparse GT operator

by $\dot{\mathbf{\Phi}}_{\mathbf{G}}(\mathcal{T}, \mathbf{X}) : X_N \to \mathbb{R}^D$. Sparse GT reduces computational complexity from quadratic in the $N$ to quadratic in the worst-case $k$-hop neighborhood cardinality. Additionally, the transferability results of Theorem 1 and Corollary 2 can be shown to hold equivalently for (13), by showing it converges to a restricted manifold transformer:

$$\dot{\mathbf{\Phi}}_{\mathcal{M}}(\mathcal{T}, f)(x) = \frac{\int_{\mathcal{M}} \dot{\gamma}_{xy} \mathbf{V} f(y) d\mu(y)}{\int_{\mathcal{M}} \dot{\gamma}_{xy} d\mu(y)}, \tag{14}$$

where $\dot{\gamma}_{xy} = \mathbb{1}_{B_r(x)} \exp\{\langle \mathbf{Q} f(x), \mathbf{K} f(y) \rangle\}$, and $B_r(x)$ denotes an Euclidean ball of radius $r$ centered around $x$. Analogous to the $k$-hop neighborhood, this modified kernel function restricts manifold attention contributions to a region around each point.

**Corollary 4.** *Let $X_N = \{x_i\}_{i=1}^N$ a set of points sampled from manifold $\mathcal{M}$. Let $\mathbf{\Phi}_{\mathcal{M}_i}(\mathcal{T}, f, \mathcal{L})$ denote a manifold transformer operating on manifolds $\mathcal{M}_i = \{y \in \mathcal{M} : \|x_i - y\| < r\}$. Assume that the $k$-hop neighborhood of sparse GT satisfies $\mathcal{N}^{\leq k}(x_i) \subseteq B_r(x_i)$. Then, it holds with probability $1 - \delta$,*

$$\|\dot{\mathbf{\Phi}}_{\mathbf{G}}(\mathcal{T}, \mathbf{X})(x_i) - \dot{\mathbf{\Phi}}_{\mathcal{M}}(\mathcal{T}, f)(x_i)\| \leq \Delta_{\Phi(\mathbf{X})}. \tag{15}$$

*Furthermore, sparse graph transformer with RPEARL positional encodings is size transferable with the same bound as Corollary 2.*

The proof of Corollary 4 relies on the same argument as Theorem 1, and is available in Appendix (7.1).

# 4. Experiments

## 4.1. Transferability on graph datasets

Corollary 2 implies that the performance gap of GTs versus the ideal manifold transformer should decay as graph sizes increase, a consequence that we now turn to validating empirically. From the dataset, we subsample training graphs $G_{\text{TR}}$ with sizes $N_{\text{TR}}$ taken over fractions $\alpha = \{0.05, 0.1, \dots, 1.0\}$, and evaluate on a large test graph $G_{\text{TST}}$ with size $N_{\text{TST}} \gg N_{\text{TR}}$. Our theory predicts that as $N_{\text{TR}}$ increases, the performance of GTs on $G_{\text{TST}}$ approximates that of a GT trained on the full graph.

**Datasets.** We evaluate on four node classification datasets. Here we present SNAP-Patents (Lim et al., 2021) and ArXiv-year (Lim et al., 2021). Results for OGBN-MAG (Wang et al., 2020) and REDDIT-BINARY (Yanardag & Vishwanathan, 2015) are available in Appendix 8.1, which show similar transferability patterns in cases where GCN and GT's accuracy is comparable. We note that due to the small size of Reddit's graphs (430 nodes per graph), the smallest meaningful downsample is 50% of the average graph size.

**Models.** We analyze GT in two settings: Dense GT + RPEARL (DGT) (Equation (1) and Sparse GT + RPEARL (SGT)(Equation (13)). In both cases, we implement multiheaded, scaled dot-product self-attention (Vaswani et al., 2017) with feedforward layers. RPEARL's GNNs are implemented as in Equation (2) using TAGConv filters (Du et al., 2018). In Appendix 8.1 we provide extended results comparing GNN, Exphormer (Shirzad et al., 2023) and an MLP baseline.

**GTs with RPEARL encodings exhibit transferability properties.** Figure 4b shows the test performance of GTs with increasing training fractions on full testing datasets. DGT and SGTs accuracy with a fraction of the training nodes is comparable to their peak accuracy with the largest training fraction, indicating successful transferability. For instance, SGT achieves close to 90% of its peak test accuracy with training graphs of 10.5% of the size. We observe similar patterns on MAG, Reddit and SNAP.

**GT's performance generalizes across various sizes of test set.** The heatmaps of Figure 7 show the accuracy of DGT and SGT on SNAP with increasing training fractions along the rows and increasing fractions of test graphs along the columns. In each row we can appreciate that test performance slightly increases until it stabilizes close to peak performance. This supports the claim that GTs generalize to graph sizes different to those seen during training.

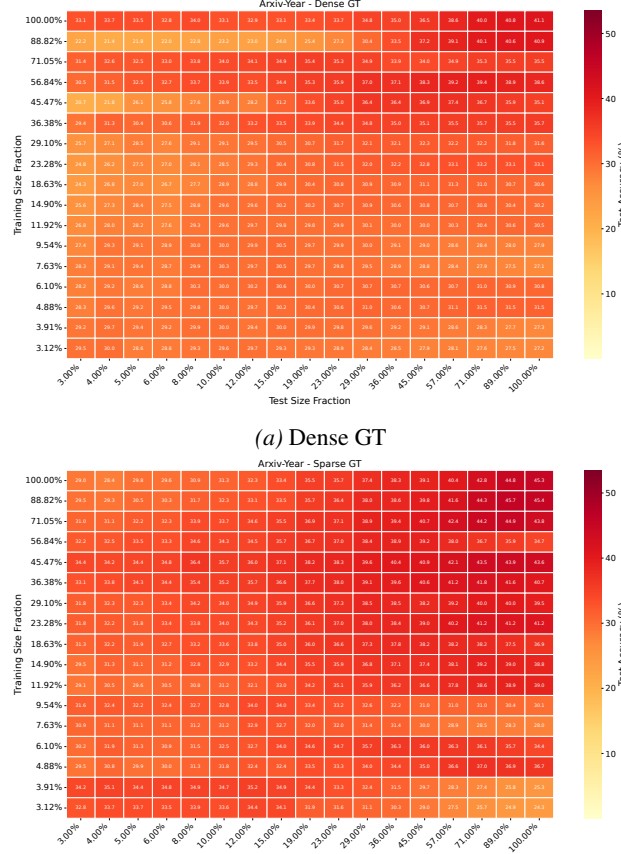

*(a)* Dense GT

*(b)* Sparse GT

*Figure 3.* Transferability heatmaps on Arxiv-year. The $x$ axis correspond to train graph sizes as a proportion of the largest graph, and the $y$ axis are test graph size fractions. The color corresponds to the test accuracy at each setting.

## 4.2. Ablation analysis and effect of masking

The $k$-hop neighborhood mask of SGT trades off computational complexity and global information, resulting up to 100x training speedups relative to DGT, while retaining comparable performance (see Table 4 in Appendix 8.2). Beyond this tradeoff, we observe improved test performance in two datasets (SNAP,ArXiv-year). We perform an ablation analysis of SGT's structural components on the SNAP Dataset to gain further insight of the effect of each mechanism on its transferability. For this, we train each architecture on a 30% downsample (513K nodes) of the largest training set, and evaluate on a 1.71M node test set. In addition to ablating RPEARL and Mask, we also consider random edges (RE) using random expander graphs of degree 3 (Shirzad et al., 2023).

**RPEARL + Masking achieves the best transference on SNAP.** On Table 1, we can observe that RPEARL encodings enhance the baseline GT performance by 10.53%.

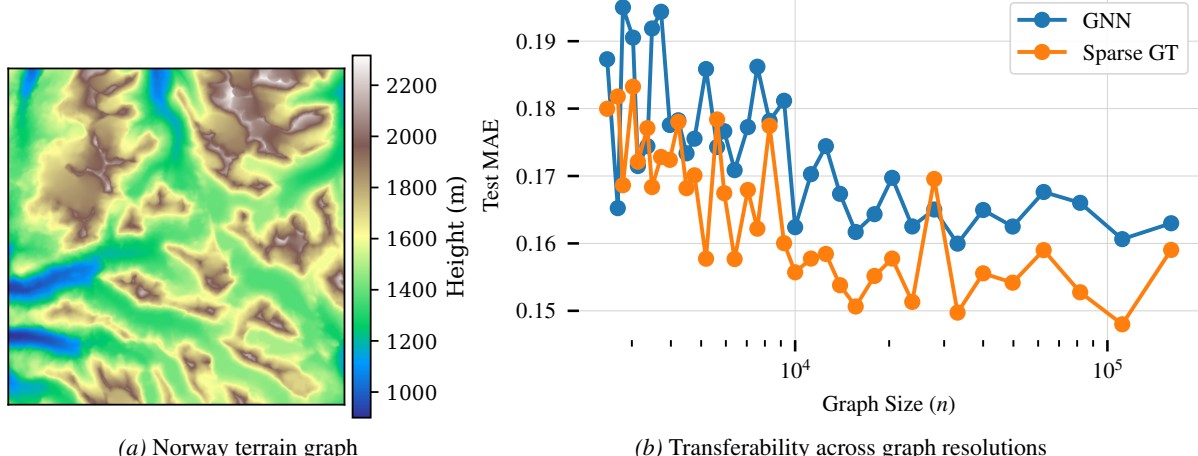

*(a)* Norway terrain graph

*(b)* Transferability across graph resolutions

*Figure 4.* Norway transferability. (a) Visualization of the norway terrain graph in full resolution. (b) Transferability plot: $x$ axis is training graph size, $y$ axis is Test MAE on the full resolution graph ($250 \times 250$).

*Table 1.* Test accuracy of GT trained with $\alpha = 0.3$, with different components in Snap-patents (1.71M nodes).

| Architecture | Accuracy | % vs. GT |
|---|---|---|
| GT (no PE) | 31.33 | – |
| GT + RPEARL | 34.63 | +10.53% |
| GT + Mask | 39.69 | +26.68% |
| GT + Mask + RPEARL | **49.70** | **+58.63%** |
| GT + Mask + RE | 31.01 | -1.02% |
| GT + Mask + RPEARL + RE | 46.55 | +48.58% |

Adding the sparse encoding mask improves performance by 26.68%. Combining both encodings yields a performance gain of 58.63%, reaching the best performance across all settings. Finally, random edges do not seem to improve performance. Our findings coincide with observations from previous works that this attention mask may be a useful inductive bias (Lin et al., 2024).

### 4.3. Terrain graph application

Terrains graphs are point clouds that approximate the manifold of a physical terrain. Some time critical applications require low latency Shortest Path Distance (SPD) estimations between two points on a terrain, which is a challenging task given that high-resolution point clouds can easily extend to millions of nodes. This has been tackled in the deep learning as a *metric learning* problem, where the task is to learn a latent space using a GNN where embedding distance approximates the true SPD between them (Chen et al., 2025).

We empirically verify the transferability of Sparse GT on the SPD approximation problem by building on the setup of (Chen et al., 2025). Each dataset is comprised of high-

resolution grid graph $\mathbf{G}$. We downsample $\mathbf{G}$ into coarser grid graphs $\mathbf{G}_r$ using strides $r \in \{4, 5, \ldots, 40\}$. For each $\mathbf{G}_r$, we train SGT with an MSE loss on the $L_1$ approximation of node embeddings against its ground truth SPDs. After training, we evaluate SGT on the full resolution $\mathbf{G}$ on a new set of sample points. For hyperparameters and more implementation details, see Appendix 8.3.

We present the results on the terrain dataset of the Troms region of Norway (Kartverket (Norwegian Mapping Authority), 2025) in Figure 4b and provide a GNN baseline for reference. We present the SPD estimation error measured by test MAE against the highest resolution $\mathbf{G}$. We observe that both architectures show strong transferability, achieving low errors on all fractions. In particular, we observe that SGT trained with $10,000$ nodes show a comparable performance than training with an order of magnitude larger graphs. We also observe that, in most fractions, Sparse GT is comparable or slightly improves the GNN performance.

## 5. Conclusions

We have presented a framework for showing that graph transformers doted with transferable positional encodings inherit their transferability. We have also presented experiments empirically verifying that various graph transformers exhibit competitive performance on large graphs when trained on smaller graphs. These results are valuable for applications dealing with large graphs, where data collection and training costs are a major challenge. This notion of transferability inheritance can be applied to other graph limit objects, for example random graphs sampled from graphons. Furthermore, other structural, relative and absolute graph positional encodings could be proven to be transferable if they satisfy the conditions of Assumption 3.

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

## 6. Appendix

### 6.1. Manifold neural networks and GNN convergence

**Laplace-Beltrami operator and manifold convolutions.** The manifold with probability density function $\rho$ is equipped with a weighted Laplace operator (Grigor'yan, 2006), generalizing the Laplace-Beltrami operator as

$$\mathcal{L}f = -\frac{1}{2\rho}\mathrm{div}(\rho^2\nabla f), \tag{16}$$

with div denoting the divergence operator of $\mathcal{M}$ and $\nabla$ denoting the gradient operator of $\mathcal{M}$ (Bronstein et al., 2017; Gross & Meinrenken, 2023). The manifold convolution operation is defined relying on the Laplace operator $\mathcal{L}$ (Wang et al., 2024c). For a function $f \in L^2(\mu)$ as input, a manifold convolutional filter (Wang et al., 2024c) can be defined as

$$g(x) = \mathbf{h}(\mathcal{L})f(x) = \sum_{k=0}^{K-1} h_k e^{-k\mathcal{L}} f(x), \tag{17}$$

with $h_k \in \mathbb{R}$ the filter parameter.

**Manifold neural networks.** A manifold neural network (MNN) is constructed by cascading $L$ layers, each of which contains a bank of manifold convolutional filters and a pointwise nonlinearity $\sigma : \mathbb{R} \to \mathbb{R}$. The output manifold function of each layer $l = 1, 2 \cdots, L$ can be explicitly denoted as

$$f_l^p(x) = \sigma\left(\sum_{q=1}^{F_{l-1}} \mathbf{h}_l^{pq}(\mathcal{L})f_{l-1}^q(x)\right), \tag{18}$$

where $f_{l-1}^q$, $1 \le q \le F_{l-1}$ is the $q$-th input feature from layer $l-1$ and $f_l^p$, $1 \le p \le F_l$ is the $p$-th output feature of layer $l$. We denote MNN as a mapping $\mathbf{\Psi}_\mathcal{M}(\mathcal{H}, \mathcal{L}, f)$, where $\mathcal{H} = \{\mathbf{h}_l^{pq}\}_{l,p,q}$ is a collective set of filter parameters in all the manifold convolutional filters.

### 6.2. Proof of GNN to MNN convergence

We first import the spectral point-wise convergence of graph Laplacian to Laplace-Beltrami operator from (Dunson et al., 2021). The spectral representation of manifold filters are similar to graph convolutional filter, while we consider the case in which the Laplace operator is self-adjoint, positive-semidefinite and the manifold $\mathcal{M}$ is compact. In this case, $\mathcal{L}_\rho$ has real, positive and discrete eigenvalues $\{\lambda_i\}_{i=1}^\infty$, written as $\mathcal{L}_\rho \phi_i = \lambda_i \phi_i$ where $\phi_i$ is the eigenfunction associated with eigenvalue $\lambda_i$. The eigenvalues are ordered in increasing order as $0 = \lambda_1 \le \lambda_2 \le \lambda_3 \le \ldots$, and the eigenfunctions are orthonormal and form an eigenbasis of $L^2(\mu)$. When mapping a manifold signal onto the eigenbasis $[\hat{f}]_i = \langle f, \phi_i \rangle_\mathcal{M} = \int_\mathcal{M} f(x)\phi_i(x)\mathrm{d}\mu(x)$, the manifold convolution can be seen in the spectral domain as

$$[\hat{g}]_i = \sum_{k=0}^{K-1} h_k e^{-k\lambda_i}[\hat{f}]_i. \tag{19}$$

Hence, the frequency response of manifold filter is given by $\hat{h}(\lambda) = \sum_{k=0}^{K-1} h_k e^{-k\lambda}$.

**Proposition 1.** *(Dunson et al., 2021)[Theorem 4] For a sufficiently small $\epsilon > 0$, if $n$ is sufficiently large so that $\epsilon = \epsilon(n) \ge \left(\frac{\log n}{n}\right)^{\frac{1}{2d+12}}$, then with probability greater than $1 - n^{-2}$, for all $0 \le i < M$,*

$$|\lambda_{i,N} - \lambda_i| \le \Omega_1\epsilon^2, \max_{x_j \in X_N} |a_i[\phi_{i,n}]_j - \phi_i(x_j)| \le \Omega_2\epsilon^2, \tag{20}$$

*with $\Omega_1$ and $\Omega_2$ related to the eigengap of $\mathcal{L}$, $d$, and the diameter, the volume, the injectivity radius, the curvature and the second fundamental form of the manifold.*

Because $\{x_1, x_2, \cdots, x_N\}$ is a set of randomly sampled points from $\mathcal{M}$, based on Theorem 19 in (Von Luxburg et al., 2008) we can claim that

$$|\langle \mathbf{S}_N f, \mathbf{S}_N \phi_i \rangle - \langle f, \phi_i \rangle_\mathcal{M}| = O\left(\sqrt{\frac{\log(1/\delta)}{N}}\right), \tag{21}$$

where $\langle f, \phi_i \rangle = \int_{\mathcal{M}} f(x) \phi_i(x) \mathrm{d}\mu(x)$ is defined as the inner product over manifold $\mathcal{M}$. This also indicates that

$$\left| \|\mathbf{S}_N f\|^2 - \|f\|_{\mathcal{M}}^2 \right| = O\left( \sqrt{\frac{\log(1/\delta)}{N}} \right), \tag{22}$$

which indicates $\|\mathbf{S}_N f\| = \|f\|_{\mathcal{M}} + O((\log(1/\delta)/N)^{1/4})$, where $\|f\|_{\mathcal{M}}^2 = \langle f, f \rangle_{\mathcal{M}}$. We suppose that the input manifold signal is $\lambda_M$-bandlimited with $M$ spectral components. We first write out the difference on each node $x_j \in X_N$ as

$$\|[\mathbf{h}(\mathbf{L}_N) \mathbf{S}_N f]_j - (\mathbf{h}(\mathcal{L}_\rho) f)(x_j)\| = \left\| \sum_{i=1}^{N} \hat{h}(\lambda_{i,N}) \langle \mathbf{S}_N f, \phi_{i,N} \rangle [\phi_{i,N}]_j - \sum_{i=1}^{M} \hat{h}(\lambda_i) \langle f, \phi_i \rangle_{\mathcal{M}} \phi_i(x_j) \right\| \tag{23}$$

$$\leq \left\| \sum_{i=1}^{M} \hat{h}(\lambda_{i,N}) \langle \mathbf{S}_N f, \phi_{i,N} \rangle [\phi_{i,N}]_j - \sum_{i=1}^{M} \hat{h}(\lambda_i) \langle f, \phi_i \rangle_{\mathcal{M}} \phi_i(x_j) \right.$$

$$\left. + \sum_{i=M+1}^{N} \hat{h}(\lambda_{i,N}) \langle \mathbf{S}_N f, \phi_{i,N} \rangle [\phi_{i,N}]_j \right\| \tag{24}$$

$$\leq \left\| \sum_{i=1}^{M} \hat{h}(\lambda_{i,N}) \langle \mathbf{S}_N f, \phi_{i,N} \rangle [\phi_{i,N}]_j - \sum_{i=1}^{M} \hat{h}(\lambda_i) \langle f, \phi_i \rangle_{\mathcal{M}} \phi_i(x_j) \right\|$$

$$+ \left\| \sum_{i=M+1}^{N} \hat{h}(\lambda_{i,N}) \langle \mathbf{S}_N f, \phi_{i,N} \rangle [\phi_{i,N}]_j \right\|. \tag{25}$$

The first part of (25) can be decomposed with the triangle inequality as follows

$$\left\| \sum_{i=1}^{M} \hat{h}(\lambda_{i,N}) \langle \mathbf{S}_N f, \phi_{i,N} \rangle [\phi_{i,N}]_j - \sum_{i=1}^{M} \hat{h}(\lambda_i) \langle f, \phi_i \rangle_{\mathcal{M}} \phi_i(x_j) \right\|$$

$$\leq \left\| \sum_{i=1}^{M} \left( \hat{h}(\lambda_{i,N}) - \hat{h}(\lambda_i) \right) \langle \mathbf{S}_N f, \phi_{i,N} \rangle [\phi_{i,N}]_j \right\| + \left\| \sum_{i=1}^{M} \hat{h}(\lambda_i) \left( \langle \mathbf{S}_N f, \phi_{i,N} \rangle [\phi_{i,N}]_j - \langle f, \phi_i \rangle_{\mathcal{M}} \phi_i(x_j) \right) \right\|. \tag{26}$$

In (26), the first part relies on the difference of eigenvalues and the second part depends on the eigenvector difference. The first term in (26) is bounded with Cauchy-Schwartz inequality as

$$\left\| \sum_{i=1}^{M} (\hat{h}(\lambda_{i,n}) - \hat{h}(\lambda_i)) \langle \mathbf{S}_N f, \phi_{i,N} \rangle [\phi_{i,N}]_j \right\| \leq \sum_{i=1}^{M} \left| \hat{h}(\lambda_{i,N}) - \hat{h}(\lambda_i) \right| |\langle \mathbf{S}_N f, \phi_{i,N} \rangle| \tag{27}$$

$$\leq \|\mathbf{S}_N f\| \sum_{i=1}^{M} |\hat{h}'(\lambda_i)| |\lambda_{i,N} - \lambda_i|$$

$$\leq \|\mathbf{S}_N f\| \sum_{i=1}^{M} C_L \Omega_1 \epsilon^2 \lambda_i^{-d} \tag{28}$$

$$\leq \|\mathbf{S}_N f\| C_L \Omega_1 \epsilon^2 \sum_{i=1}^{M} i^{-2} \tag{29}$$

$$\leq \left( \|f\|_{\mathcal{M}} + \left( \frac{\log(1/\delta)}{N} \right)^{\frac{1}{4}} \right) \Omega_1 \epsilon^2 \frac{\pi^2}{6} := A_1(N) \tag{30}$$

In (27), it depends on the inequality that $|[\phi_{i,N}]_j| \leq \|\phi_{i,N}\|_\infty \leq \|\phi_{i,N}\|_2 = 1$. In (28), it depends on the filter continuity assumption. In (29), we implement Weyl's law (Arendt et al., 2009) which indicates that eigenvalues of Laplace operator scales with the order $\lambda_i \sim i^{2/d}$. The last inequality comes from the fact that $\sum_{i=1}^{\infty} i^{-2} = \frac{\pi^2}{6}$. The second term in (26) can

be bounded with the triangle inequality as

$$\left\| \sum_{i=1}^{M} \hat{h}(\lambda_i) \left( \langle \mathbf{S}_N f, \phi_{i,N} \rangle [\phi_{i,N}]_j - \langle f, \phi_i \rangle_{\mathcal{M}} \phi_i(x_j) \right) \right\|$$

$$\leq \left\| \sum_{i=1}^{M} \hat{h}(\lambda_i) \left( \langle \mathbf{S}_N f, \phi_{i,N} \rangle [\phi_{i,N}]_j - \langle \mathbf{S}_N f, \phi_{i,N} \rangle \phi_i(x_j) \right) \right\| + \left\| \sum_{i=1}^{M} \hat{h}(\lambda_i) \left( \langle \mathbf{S}_N f, \phi_{i,N} \rangle \phi_i(x_j) - \langle f, \phi_i \rangle_{\mathcal{M}} \phi_i(x_j) \right) \right\| \tag{31}$$

The first term in (31) can be bounded with inserting the eigenfunction convergence result in Proposition 1 as

$$\left\| \sum_{i=1}^{M} \hat{h}(\lambda_i) \left( \langle \mathbf{S}_N f, \phi_{i,N} \rangle [\phi_{i,N}]_j - \langle \mathbf{S}_N f, \phi_{i,N} \rangle_{\mathcal{M}} \phi_i(x_j) \right) \right\|$$

$$\leq \sum_{i=1}^{M} \left| \hat{h}(\lambda_i) \right| \|\mathbf{S}_N f\| \, |[\phi_{i,N}]_j - \phi_i(x_j)| \tag{32}$$

$$\leq \sum_{i=1}^{M} (\lambda_i^{-d+1}) \Omega_2 \epsilon^2 \left( \|f\|_{\mathcal{M}} + \left( \frac{\log(1/\delta)}{N} \right)^{\frac{1}{4}} \right) \tag{33}$$

$$\leq \Omega_2 \epsilon^2 \sum_{i=1}^{M} (\lambda_i^{-d+1}) \left( \|f\|_{\mathcal{M}} + \left( \frac{\log(1/\delta)}{N} \right)^{\frac{1}{4}} \right) \tag{34}$$

$$:= A_2(M, N). \tag{35}$$

Considering the filter continuity assumption, the second term in (31) can be written as

$$\left\| \sum_{i=1}^{M} \hat{h}(\lambda_{i,N}) (\langle \mathbf{S}_N f, \phi_{i,N} \rangle \phi_i(x_j) - \langle f, \phi_i \rangle_{\mathcal{M}} \phi_i(x_j)) \right\|$$

$$\leq \sum_{i=1}^{M} \left| \hat{h}(\lambda_{i,N}) \right| |\langle \mathbf{S}_N f, \phi_{i,N} \rangle - \langle f, \phi_i \rangle_{\mathcal{M}}| \, |\phi_i(x_j)| \tag{36}$$

$$\leq \sum_{i=1}^{M} (\lambda_{i,N}^{-d}) |\langle \mathbf{S}_N f, \phi_{i,N} \rangle - \langle f, \phi_i \rangle_{\mathcal{M}}| \, \|\phi_i\| \tag{37}$$

$$\leq \sum_{i=1}^{M} \left( 1 + \Omega_1 \epsilon^2 \right)^{-d} \lambda_i^{-d} |\langle \mathbf{S}_N f, \phi_{i,N} \rangle - \langle f, \phi_i \rangle_{\mathcal{M}}| \tag{38}$$

$$\leq \frac{\pi^2}{6} |\langle \mathbf{S}_N f, \phi_{i,N} \rangle - \langle f, \phi_i \rangle_{\mathcal{M}}| := A_3(N) \tag{39}$$

The term $|\langle \mathbf{S}_N f, \phi_{i,N} \rangle - \langle f, \phi_i \rangle_{\mathcal{M}}|$ can be decomposed by inserting a term $\langle \mathbf{S}_N f, \mathbf{S}_N \phi_i \rangle$ as

$$|\langle \mathbf{S}_N f, \phi_{i,N} \rangle - \langle f, \phi_i \rangle_{\mathcal{M}}| \leq |\langle \mathbf{S}_N f, \phi_{i,N} \rangle - \langle \mathbf{S}_N f, \mathbf{S}_N \phi_i \rangle + \langle \mathbf{S}_N f, \mathbf{S}_N \phi_i \rangle - \langle f, \phi_i \rangle_{\mathcal{M}}| \tag{40}$$

$$\leq |\langle \mathbf{S}_N f, \phi_{i,N} \rangle - \langle \mathbf{S}_N f, \mathbf{S}_N \phi_i \rangle| + |\langle \mathbf{S}_N f, \mathbf{S}_N \phi_i \rangle - \langle f, \phi_i \rangle_{\mathcal{M}}| \tag{41}$$

$$\leq \|\mathbf{S}_N f\| \|\phi_{i,N} - \mathbf{S}_N \phi_i\| + |\langle \mathbf{S}_N f, \mathbf{S}_N \phi_i \rangle - \langle f, \phi_i \rangle_{\mathcal{M}}| \tag{42}$$

$$\leq \left( \|f\|_{\mathcal{M}} + \left( \frac{\log(1/\delta)}{N} \right)^{\frac{1}{4}} \right) \frac{C_{\mathcal{M},2} \lambda_i \sqrt{\epsilon}}{\theta_i} + \sqrt{\frac{\log(1/\delta)}{N}} \tag{43}$$

Then equation (38) can be bounded as

$$\left\| \sum_{i=1}^{M} \hat{h}(\lambda_{i,N})(\langle \mathbf{S}_N f, \phi_{i,N} \rangle \phi_i(x_j) - \langle f, \phi_i \rangle_{\mathcal{M}} \phi_i(x_j)) \right\|$$

$$\leq \sum_{i=1}^{M} (1 + \Omega_1 \epsilon^2)^{-d} (\lambda_i^{-d}) \left( \left( \|f\|_{\mathcal{M}} + \left( \frac{\log(1/\delta)}{N} \right)^{\frac{1}{4}} \right) \frac{C_{\mathcal{M},2} \lambda_i \epsilon}{\theta_i} + \sqrt{\frac{\log(1/\delta)}{N}} \right) \tag{44}$$

$$\leq \frac{\pi^2}{6} \max_{i=1,\cdots,M} \frac{C_{\mathcal{M},2} \epsilon}{\theta_i} \left( \|f\|_{\mathcal{M}} + \left( \frac{\log(1/\delta)}{N} \right)^{\frac{1}{4}} \right) + \frac{\pi^2}{6} \sqrt{\frac{\log(1/\delta)}{N}} \tag{45}$$

The second term in (25) can be bounded with the eigenvalue difference bound in Proposition 1 as

$$\left\| \sum_{i=M+1}^{N} \hat{h}(\lambda_{i,N}) \langle \mathbf{S}_N f, \phi_{i,N} \rangle [\phi_{i,N}]_j \right\| \leq \sum_{i=M+1}^{N} (\lambda_{i,N}^{-d}) \left( \|f\|_{\mathcal{M}} + \left( \frac{\log(1/\delta)}{N} \right)^{\frac{1}{4}} \right) \tag{46}$$

$$\leq \sum_{i=M+1}^{\infty} (\lambda_{i,N}^{-d}) \|f\|_{\mathcal{M}} \tag{47}$$

$$\leq \left( 1 + \Omega_1 \epsilon^2 \right)^{-d} \sum_{i=M+1}^{\infty} (\lambda_i^{-d}) \|f\|_{\mathcal{M}} \tag{48}$$

$$\leq M^{-1} \|f\|_{\mathcal{M}} := A_4(M). \tag{49}$$

We note that the bound is made up by terms $A_1(N) + A_2(M,N) + A_3(N) + A_4(M)$, related to the bandwidth of manifold signal $M$ and the number of sampled points $N$. This makes the bound scale with the order

$$\|[\mathbf{h}(\mathbf{L}_N)\mathbf{S}_N f]_j - \mathbf{h}(\mathcal{L}_\rho)f(x_j)\| \leq C_1' \epsilon^2 + C_2' \epsilon \theta_M^{-1} + C_3' \sqrt{\frac{\log(1/\delta)}{N}} + C_4' M^{-1}, \tag{50}$$

with $C_1' = C_L \Omega_1 \frac{\pi^2}{6} \|f\|_{\mathcal{M}}$, $C_2' = \Omega_2 \frac{\pi^2}{6}$, $C_3' = \frac{\pi^2}{6}$ and $C_4' = \|f\|_{\mathcal{M}}$. As $N$ goes to infinity, for every $\delta > 0$, there exists some $M_0$, such that for all $M > M_0$ it holds that $A_4(M) \leq \delta/2$. There also exists $n_0$, such that for all $N > n_0$, it holds that $A_1(N) + A_2(M_0, N) + A_3(N) \leq \delta/2$. We can conclude that the summations converge as $N$ goes to infinity. We see $M$ large enough to have $M^{-1} \leq \delta'$, which makes the eigengap $\theta_M$ also bounded by $\epsilon$. We combine the first two terms as

$$\|[\mathbf{h}(\mathbf{L}_N)\mathbf{S}_N f]_j - \mathbf{h}(\mathcal{L}_\rho)f(x_j)\| \leq (C_1 C_L + C_2)\epsilon^2 + \frac{\pi^2}{6} \sqrt{\frac{\log(1/\delta)}{N}}, \tag{51}$$

with $C_1 = \Omega_1 \frac{\pi^2}{6} \|f\|_{\mathcal{M}}$ and $C_2 = \Omega_2 \frac{\pi^2}{6} \theta_{\delta'-1}^{-1}$. To bound the output difference of MNNs, we need to write in the form of features of the final layer

$$\|[\mathbf{\Psi_G}(\mathbf{H}, \mathbf{L}_N, \mathbf{S}_N f)]_j - \mathbf{\Psi}(\mathbf{H}, \mathcal{L}_\rho, f)(x_j)\| = \left\| \sum_{q=1}^{F} [\mathbf{x}_{n,L}^q]_j - \sum_{q=1}^{F} f_L^q(x_j) \right\| \tag{52}$$

$$\leq \sum_{q=1}^{F} \left\| [\mathbf{x}_{n,L}^q]_j - f_L^q(x_j) \right\|. \tag{53}$$

By inserting the definitions, we have

$$\left\| [\mathbf{x}_{n,l}^p]_j - f_l^p(x_j) \right\| = \left\| \sigma\left( \left[ \sum_{q=1}^{F} \mathbf{h}_l^{pq}(\mathbf{L}_N) \mathbf{x}_{n,l-1}^q \right]_j \right) - \sigma\left( \sum_{q=1}^{F} \mathbf{h}_l^{pq}(\mathcal{L}_\rho) f_{l-1}^q(x_j) \right) \right\| \tag{54}$$

with $\mathbf{x}_{n,0} = \mathbf{S}_N f$ as the input of the first layer. With a normalized point-wise Lipschitz nonlinearity, we have

$$\|[\mathbf{x}_{n,l}^p]_j - f_l^p(x_j)\| \le \left\| \sum_{q=1}^F \left[ \mathbf{h}_l^{pq}(\mathbf{L}_N)\mathbf{x}_{n,l-1}^q \right]_j - \sum_{q=1}^F \mathbf{h}_l^{pq}(\mathcal{L}_\rho) f_{l-1}^q(x_j) \right\| \tag{55}$$

$$\le \sum_{q=1}^F \left\| [\mathbf{h}_l^{pq}(\mathbf{L}_N)\mathbf{x}_{n,l-1}^q]_j - \mathbf{h}_l^{pq}(\mathcal{L}_\rho) f_{l-1}^q(x_j) \right\| \tag{56}$$

The difference can be further decomposed as

$$\|[\mathbf{h}_l^{pq}(\mathbf{L}_N)\mathbf{x}_{n,l-1}^q]_j - \mathbf{h}_l^{pq}(\mathcal{L}_\rho) f_{l-1}^q(x_j)\|$$
$$\le \|[\mathbf{h}_l^{pq}(\mathbf{L}_N)\mathbf{x}_{n,l-1}^q]_j - [\mathbf{h}_l^{pq}(\mathbf{L}_N)\mathbf{S}_N f_{l-1}^q]_j + [\mathbf{h}_l^{pq}(\mathbf{L}_N)\mathbf{S}_N f_{l-1}^q]_j - \mathbf{h}_l^{pq}(\mathcal{L}_\rho) f_{l-1}^q(x_j)\| \tag{57}$$

$$\le \left\| [\mathbf{h}_l^{pq}(\mathbf{L}_N)\mathbf{x}_{n,l-1}^q]_j - [\mathbf{h}_l^{pq}(\mathbf{L}_N)\mathbf{S}_N f_{l-1}^q]_j \right\| + \left\| [\mathbf{h}_l^{pq}(\mathbf{L}_N)\mathbf{S}_N f_{l-1}^q]_j - \mathbf{h}_l^{pq}(\mathcal{L}_\rho) f_{l-1}^q(x_j) \right\| \tag{58}$$

The second term can be bounded with (50) and we denote the bound as $\Delta_N$ for simplicity. The first term can be decomposed by Cauchy-Schwartz inequality and non-amplifying of the filter functions as

$$\left\| [\mathbf{x}_{n,l}^p]_j - f_l^p(x_j) \right\| \le \sum_{q=1}^F \Delta_N \|\mathbf{x}_{n,l-1}^q\| + \sum_{q=1}^F \|[\mathbf{x}_{l-1}^q]_j - f_{l-1}^q(x_j)\|. \tag{59}$$

To solve this recursion, we need to compute the bound for $\|\mathbf{x}_l^p\|$. By normalized Lipschitz continuity of $\sigma$ and the fact that $\sigma(0) = 0$, we can get

$$\|\mathbf{x}_l^p\| \le \left\| \sum_{q=1}^F \mathbf{h}_l^{pq}(\mathbf{L}_N)\mathbf{x}_{l-1}^q \right\| \le \sum_{q=1}^F \|\mathbf{h}_l^{pq}(\mathbf{L}_N)\| \|\mathbf{x}_{l-1}^q\| \le \sum_{q=1}^F \|\mathbf{x}_{l-1}^q\| \le F^{l-1}\|\mathbf{x}\|. \tag{60}$$

Insert this conclusion back to solve the recursion, we can get

$$\left\| [\mathbf{x}_{n,l}^p]_j - f_l^p(x_j) \right\| \le l F^{l-1} \Delta_N \|\mathbf{x}\|. \tag{61}$$

Replace $l$ with $L$ we can obtain

$$\|[\mathbf{\Psi}_{\mathbf{G}}(\mathbf{H}, \mathbf{L}_N, \mathbf{S}_N f)]_j - \mathbf{\Psi}(\mathbf{H}, \mathcal{L}_\rho, f)(x_j)\| \le L F^{L-1} \Delta_N, \tag{62}$$

when the input graph signal is normalized. By replacing $f = \mathbf{I}_N \mathbf{x}$, we can conclude the proof.

### 6.3. Local Lipschitz continuity of MNNs

We utilize Proposition 3 in (Wang et al., 2024a), which shows that the outputs of MNN defined in (18) are locally Lipschitz continuous within a certain area, which is stated explicitly as follows.

**Proposition 2.** *(Local Lipschitz continuity of MNNs (Wang et al., 2024a)[Proposition 3]) Assume that the assumptions in Theorem 1 hold. Let MNN be $L$ layers with $F$ features in each layer, suppose the manifold filters are nonamplifying with $|\hat{h}(\lambda)| \le 1$ and the nonlinearities normalized Lipschitz continuous, then there exists a constant $C'$ such that*

$$|\mathbf{\Phi}(\mathbf{H}, \mathcal{L}_\rho, f)(x) - \mathbf{\Phi}(\mathbf{H}, \mathcal{L}_\rho, f)(y)| \le F^L C' dist(x - y), \quad \text{for all } x, y \in B_r(\mathcal{M}), \tag{63}$$

*where $B_r(\mathcal{M})$ is a ball with radius $r$ over $\mathcal{M}$ with respect to the geodesic distance.*

### 6.4. Manifold decomposition and induced manifold signal

The graph $\mathbf{G}$ contains points $X_N = \{x_i\}_{i=1}^N$ sampled from the manifold $\mathcal{M}$. The graph signal $\mathbf{X} \in \mathbb{R}^{N \times D}$ can be viewed as a discretization of the continuous manifold signal $f$ evaluated at points $X_N$, that is

$$\mathbf{S}_N f = \mathbf{X}, \tag{64}$$

where $\mathbf{S}_N : L^2(\mu) \to L_2(X_N)$ is called the sampling operator. The sample $X_N$ induces a decomposition of the manifold (Trillos et al., 2018), $\{V_i\}_{i=1}^N$ with respect to $X_N$, with $V_i \subset B_r(x_i)$ a ball of radius $r$ centered at $x_i$, with respect to the Euclidean distance in Euclidean ambient space.

Let $\mu_N = \frac{1}{N} \sum_{i=1}^N \delta x_i$ the empirical measure of the random sample. The decomposition is defined by the $\infty$-optimal transport map $T : \mathcal{M} \to X_N$, defined by the $\infty$-optimal Transport Distance between $\mu$ and $\mu_N$,

$$d_\infty(\mu, \mu_N)L = \min_{T:T\#\mu=\mu_N} \text{ess sup}_{x\in\mathcal{M}} \delta(x, t(x)). \tag{65}$$

Here, $T\#\mu$ denotes that $\mu(T^{-1}(V)) = \mu_N(V)$ holds for every $V_i$ of the decomposition of $\mathcal{M}$.

The radius of the balls where the partitions are contained can be bounded as $r \leq A(\frac{\log N}{N})^{1/d}$ when $d \geq 3$ and as $r \leq A(\log N)^{3/4}/N^{1/2}$ when $d = 2$, with $A$ being a constant related to the geometry of the manifold.

The manifold function induced by the signals of the sampled graph is a piecewise constant function defined by

$$(\mathbf{I}_N \mathbf{X})(x) = \sum_{i=1}^N [\mathbf{X}]_i \mathbb{1}_{x\in V_i}, \tag{66}$$

where $\mathbf{I}_N : L_2(X_N) \to L^2(\mu)$ denotes the interpolation operator.

### 6.5. Lemmas and Propositions

In this section we state a series of lemmas that will be useful for the proof of Theorem 1.

**Lemma 5.** *Let $x \in \mathcal{M}$ a point in the manifold, and $y \in V_j$, a point in partition $V_j \subset B_r(x_j)$. Then it holds that*

$$|\langle \mathbf{Q}f(x), \mathbf{K}f(x_j)\rangle - \langle \mathbf{Q}f(x), \mathbf{K}f(y)\rangle| \leq B_f C_Q C_K \, r \tag{67}$$

*Proof.*

$$|\langle \mathbf{Q}f(x), \mathbf{K}f(x_j)\rangle - \langle \mathbf{Q}f(x), \mathbf{K}f(y)\rangle| = |\langle \mathbf{Q}f(x), \mathbf{K}(f(x_j) - f(y))\rangle| \tag{68}$$
$$\leq C_Q \|f(x)\| - C_K \|f(x_j) - f(y)\| \tag{69}$$
$$\leq B_f C_Q C_K \|f(x_j) - f(y)\| \tag{70}$$
$$\leq B_f C_Q C_K |x_j - y| \tag{71}$$
$$\leq B_f C_Q C_K \, r \tag{72}$$

Where in (69) we apply the bound on the linear operators $\mathbf{Q}$ and $\mathbf{K}$, in (70) we apply the bound on the manifold signal $\|f(x)\| \leq B_f$, in (71) we apply the assumption on normalized Lipschitz MNN, $\|f(x) - f(y)\| \leq |x - y|$, and in (72) we use the fact that $y \in V_j$, therefore $|y - x_j| \leq r$. $\qquad\square$

For notational brevity, onwards we will denote the GT attention coefficients as $\gamma_{ij} = \exp[\langle \mathbf{Q}\mathbf{x}_i, \mathbf{K}\mathbf{x}_j\rangle]$, the MT attention coefficients as $\tilde{\gamma}_{iy} = \exp[\langle \mathbf{Q}f(x_i)\mathbf{K}f(y)\rangle]$, and the induced manifold signal coefficients as $\tilde{\gamma}_{ij} = \exp[\langle \mathbf{Q}f(x_i), \mathbf{K}f(x_j)\rangle]$.

**Lemma 6.** *Let $X_N = \{x_i\}_{i=1}^N$ be a set of points sampled from the manifold $\mathcal{M}$, with its corresponding induced partitioning $\{V_i\}_{i=1}^N$. For each $x_j \in X_N$, and for any $y \in V_j$, it holds that*

$$|\tilde{\gamma}_{ij} - \tilde{\gamma}_{iy}| \leq e^M B_f C_Q C_K \, r \tag{73}$$

*Proof.*

$$|\tilde{\gamma}_{ij} - \tilde{\gamma}_{iy}| = |\exp\langle \mathbf{Q}f(x_i), \mathbf{K}f(x_j)\rangle - \exp\langle \mathbf{Q}f(x_i), \mathbf{K}f(y)\rangle| \tag{74}$$
$$\leq e^M |\langle \mathbf{Q}f(x_i), \mathbf{K}f(x_j)\rangle - \langle \mathbf{Q}f(x_i), \mathbf{K}f(y)\rangle| \tag{75}$$
$$\leq e^M B_f C_Q C_K \, r \tag{76}$$

where $M := \sup_{u,v \in \mathcal{M}} \langle \mathbf{Q}f(u), \mathbf{K}f(v) \rangle$. Note that $M \leq C_{QK}B_f^2$ using the bounds on the MNN signal and linear operators. In the first inequality we use the mean value theorem, $|e^a - e^b| \leq e^{\max\{a,b\}}|a - b|$. In the second we apply the bound from Lemma 5. $\square$

**Lemma 7.** *For each $x_i, x_j \in X_N$ it holds that*

$$|\gamma_{ij} - \tilde{\gamma}_{ij}| \leq 2e^M C_{QK} \Delta_{\text{PE}}. \tag{77}$$

*Proof.*

$$|\gamma_{ij} - \tilde{\gamma}_{ij}| = \left| \exp\big(\langle \mathbf{Q}\mathbf{x}_i, \mathbf{K}\mathbf{x}_j \rangle\big) - \exp\big(\langle \mathbf{Q}f(x_i), \mathbf{K}f(x_j) \rangle\big) \right| \tag{78}$$

$$\leq e^M \left| \langle \mathbf{Q}\mathbf{x}_i, \mathbf{K}\mathbf{x}_j \rangle - \langle \mathbf{Q}f(x_i), \mathbf{K}f(x_j) \rangle \right| \tag{79}$$

$$\leq e^M \left( \left| \langle \mathbf{Q}(\mathbf{x}_i - f(x_i)), \mathbf{K}\mathbf{x}_j \rangle \right| + \left| \langle \mathbf{Q}f(x_i), \mathbf{K}(\mathbf{x}_j - f(x_j)) \rangle \right| \right) \tag{80}$$

$$\leq e^M \left( \|\mathbf{Q}\| \|\mathbf{K}\| \|\mathbf{x}_i - f(x_i)\| \|\mathbf{x}_j\| + \|\mathbf{Q}\| \|\mathbf{K}\| \|f(x_i)\| \|\mathbf{x}_j - f(x_j)\| \right) \tag{81}$$

$$\leq 2e^M C_{QK} B_f \, \Delta_{\text{PE}} \ \leq \ 2e^M C_{QK} \Delta_{\text{PE}}. \tag{82}$$

In (79) we apply the mean value theorem $|e^a - e^b| \leq e^{\max\{a,b\}}|a - b|$ and upper-bound $\max\{a, b\} \leq M$. In (80) we add and subtract $\langle \mathbf{Q}f(x_i), \mathbf{K}\mathbf{x}_j \rangle$ and use bilinearity and the triangle inequality. In (81) we apply Cauchy–Schwarz and boundedness of the linear operators. In (82) we use Assumption 3 and $B_f = 1$. $\square$

**Lemma 8.** *Let $x, y \in \mathcal{M}$, with $|x - x_i| \leq r$. Then, it holds that*

$$|\tilde{\gamma}_{iy} - \tilde{\gamma}_{xy}| \leq e^M C_{QK} B_f r \tag{83}$$

*Proof.*

$$|\tilde{\gamma}_{iy} - \tilde{\gamma}_{xy}| = |\exp\langle \mathbf{Q}f(x_i), \mathbf{K}f(y) \rangle - \exp\langle \mathbf{Q}f(x), \mathbf{K}f(y) \rangle| \tag{84}$$

$$\leq e^M |\langle \mathbf{Q}f(x_i), \mathbf{K}f(y) \rangle - \exp\langle \mathbf{Q}f(x), \mathbf{K}f(y) \rangle| \tag{85}$$

$$= e^M |\langle \mathbf{Q}\left[f(x_i) - f(x)\right], \mathbf{K}f(y) \rangle| \tag{86}$$

$$\leq e^M C_{QK} B_f r. \tag{87}$$

where in (85) we use $|e^a - e^b| \leq e^{\max\{a,b\}}|a - b|$ and upper-bound $\max\{a, b\} \leq M$, in (86) we rearrange the terms, and in (87) we use the bounds on linear operators, bounded manifold signal, normalized Lipschitz of $f$, and $|x - x_i| \leq r$. $\square$

### 6.6. Proof of Theorem 1

Theorem 1 bounds the convergence of a Graph Transformer with GNN-based PEs to a Manifold Transformer with MNN-based PEs.

*Proof.* The graph transformer's output for the $i$-th node can be written in vector form as

$$\mathbf{x}_i = \frac{\sum_{j=1}^{n} \exp[\langle \mathbf{Q}\mathbf{x}_i, \mathbf{K}\mathbf{x}_j \rangle] \mathbf{V}\mathbf{x}_j}{\sum_{j=1}^{n} \exp[\langle \mathbf{Q}\mathbf{x}_i, \mathbf{K}\mathbf{x}_j \rangle]} \tag{88}$$

For notational brevity, we will denote the GT attention coefficients as $\gamma_{ij} = \exp[\langle \mathbf{Q}\mathbf{x}_i, \mathbf{K}\mathbf{x}_j \rangle]$, the MT attention coefficients as $\tilde{\gamma}_{iy} = \exp[\langle \mathbf{Q}f(x_i)\mathbf{K}f(y) \rangle]$ , and the induced manifold signal coefficients as $\tilde{\gamma}_{ij} = \exp[\langle \mathbf{Q}f(x_i), \mathbf{K}f(x_j) \rangle]$. Furthermore, when necessary we will abbreviate the denominator terms as $D_{\mathbf{G}} = \sum_{j=1}^{N} \gamma_{ij}$, $D_{\mathcal{M}} = \int_{\mathcal{M}} \tilde{\gamma}_{iy} d\mu(y)$, and $\bar{D}_{\mathcal{M}} = \sum_{j=1}^{n} \tilde{\gamma}_{ij}$.

Thus we have

$$\mathbf{\Phi_G}(\mathbf{X};\mathbf{T})(x_i) = \frac{1}{D_\mathbf{G}} \sum_{j=1}^{n} \gamma_{ij} \mathbf{V}\mathbf{x}_j \tag{89}$$

$$\mathbf{\Phi_M}(f;\mathbf{T})(x_i) = \frac{1}{D_\mathcal{M}} \int_\mathcal{M} \tilde{\gamma}_{iy} \mathbf{V} f(y) d\mu(y) \tag{90}$$

We will introduce an auxiliary term derived from the induced manifold signal, denoted by

$$\bar{\Phi}_\mathcal{M}(f;\mathbf{T})(x_i) = = \frac{\sum_{j=1}^{n} \int_{V_j} \tilde{\gamma}_{ij} \mathbf{V}\mathbf{x}_j d\mu(y)}{\sum_{j=1}^{n} \int_{V_j} \tilde{\gamma}_{ij} d\mu(y)}. \tag{91}$$

Notice that since the integrand does not depend on $y$, we can simplify $\bar{\Phi}_\mathcal{M}(f;\mathbf{T})(x_i)$ to:

$$\bar{\Phi}_\mathcal{M}(f;\mathbf{T})(x_i) = \frac{\sum_{j=1}^{n} \int_{V_j} \tilde{\gamma}_{ij} \mathbf{V}\mathbf{x}_j d\mu(y)}{\sum_{j=1}^{n} \int_{V_j} \tilde{\gamma}_{ij} d\mu(y)} \tag{92}$$

$$= \frac{\sum_{j=1}^{n} \tilde{\gamma}_{ij} \mathbf{V}\mathbf{x}_j \cdot \mu(V_j)}{\sum_{j=1}^{n} \tilde{\gamma}_{ij} \cdot \mu(V_j)} \tag{93}$$

$$= \frac{\frac{1}{N} \sum_{j=1}^{n} \tilde{\gamma}_{ij} \mathbf{V}\mathbf{x}_j}{\frac{1}{N} \sum_{j=1}^{n} \tilde{\gamma}_{ij}} \tag{94}$$

$$= \frac{1}{\bar{D}_\mathcal{M}} \sum_{j=1}^{n} \tilde{\gamma}_{ij} \mathbf{V}\mathbf{x}_j, \tag{95}$$

The output difference for node $i$ can be decomposed as

$$\|\mathbf{\Phi}_G(\mathbf{X};\mathbf{T})(x_i) - \mathbf{\Phi}_\mathcal{M}(f;\mathbf{T})(x_i)\|$$
$$= \|\mathbf{\Phi_G}(\mathbf{X};\mathbf{T})(x_i) - \bar{\Phi}_\mathcal{M}(f;\mathbf{T})(x_i) + \bar{\Phi}_\mathcal{M}(f;\mathbf{T})(x_i) - \mathbf{\Phi}_\mathcal{M}(f;\mathbf{T})(x_i)\| \tag{96}$$
$$\le \|\mathbf{\Phi_G}(\mathbf{X};\mathbf{T})(x_i) - \bar{\Phi}_\mathcal{M}(\mathbf{X};\mathbf{T})(x_i)\| + \|\bar{\Phi}_\mathcal{M}(\mathbf{X};\mathbf{T})(x_i) - \mathbf{\Phi}_\mathcal{M}(f,\mathcal{L};\mathbf{T})(x_i)\| \tag{97}$$

To Equation (96) we add and subtract the induced manifold signal term, and in (96) to (97) we use the triangle inequality. We will now bound the first term of (97),

$$\left|\mathbf{\Phi_G}(\mathbf{X};\mathbf{T})(x_i) - \bar{\Phi}_\mathcal{M}(\mathbf{X};\mathbf{T})(x_i)\right| = \left|\frac{1}{D_\mathbf{G}} \sum_{j=1}^{n} \gamma_{ij} \mathbf{V}\mathbf{x}_j - \frac{1}{\bar{D}_\mathcal{M}} \sum_{j=1}^{n} \tilde{\gamma}_{ij} \mathbf{V} f(x_j)\right|. \tag{98}$$

Distribute the denominators, add and subtract $D_\mathcal{M} \gamma_{ij} \mathbf{V} f(x_j)$, then apply triangle inequality:

$$\left\|\frac{1}{D_\mathbf{G}\bar{D}_\mathcal{M}} \sum_{j=1}^{N} \bar{D}_\mathcal{M}\gamma_{ij} \mathbf{V}\mathbf{x}_j - D_\mathbf{G}\tilde{\gamma}_{ij} \mathbf{V} f(x_j)\right\|$$

$$= \left\|\frac{1}{D_\mathbf{G}\bar{D}_\mathcal{M}} \sum_{j=1}^{N} \bar{D}_\mathcal{M}\gamma_{ij} \mathbf{V}\mathbf{x}_j - \bar{D}_\mathcal{M}\gamma_{ij} \mathbf{V} f(x_j) + \bar{D}_\mathcal{M}\gamma_{ij} \mathbf{V} f(x_j) - D_\mathbf{G}\tilde{\gamma}_{ij} \mathbf{V} f(x_j)\right\| \tag{99}$$

$$\le \left\|\frac{1}{D_\mathbf{G}\bar{D}_\mathcal{M}} \sum_{j=1}^{N} \bar{D}_\mathcal{M}\gamma_{ij} \mathbf{V}\mathbf{x}_j - \bar{D}_\mathcal{M}\gamma_{ij} \mathbf{V} f(x_j)\right\| + \left\|\frac{1}{D_\mathbf{G}\bar{D}_\mathcal{M}} \sum_{j=1}^{N} \bar{D}_\mathcal{M}\gamma_{ij} \mathbf{V} f(x_j) - D_\mathbf{G}\tilde{\gamma}_{ij} \mathbf{V} f(x_j)\right\| \tag{100}$$

$$= \left\|\frac{1}{D_\mathbf{G}} \sum_{j=1}^{N} \gamma_{ij} \mathbf{V}(\mathbf{x}_j - f(x_j))\right\| + \left\|\frac{1}{D_\mathbf{G}\bar{D}_\mathcal{M}} \sum_{j=1}^{N} (\bar{D}_\mathcal{M}\gamma_{ij} - D_\mathbf{G}\tilde{\gamma}_{ij}) \mathbf{V} f(x_j)\right\|. \tag{101}$$

Note that $\mathbf{x}_j - f(x_j)$ represents the output difference between a GNN and an MNN evaluated at node $x_j$. By Assumption 3, this difference is bounded by $\Delta_{\mathrm{PE}}$. Using this bound, we can control the first term of (101) as follows:

$$\frac{1}{D_{\mathbf{G}}} \left\| \sum_{j=1}^{N} \gamma_{ij} \mathbf{V}(\mathbf{x}_j - f(x_j)) \right\| \leq \frac{1}{D_{\mathbf{G}}} \sum_{j=1}^{N} \gamma_{ij} \|\mathbf{V}\| \cdot \|\mathbf{x}_j - f(x_j)\| \leq \frac{C_V \Delta_{\mathrm{PE}}}{D_{\mathbf{G}}} \sum_{j=1}^{N} \gamma_{ij} = C_V \Delta_{\mathrm{PE}}, \qquad (102)$$

where we first apply the triangle inequality, along with the definition of the operator norm. Finally (102) uses the bound on the operator norm of $\mathbf{V}$ and $\|\mathbf{x}_j - f(x_j)\| \leq \Delta_{\mathrm{PE}}$ from Assumption 3, followed by the identity $\sum_{j=1}^{N} \gamma_{ij} = D_{\mathbf{G}}$.

Now we bound the second term in (101). We add and subtract another comparative term $\bar{D}_{\mathcal{M}} \tilde{\gamma}_{ij}$,

$$\left\| \frac{1}{D_{\mathbf{G}} \bar{D}_{\mathcal{M}}} \sum_{j=1}^{N} (\bar{D}_{\mathcal{M}} \gamma_{ij} - D_{\mathbf{G}} \tilde{\gamma}_{ij}) \mathbf{V} f(x_j) \right\|$$

$$= \left\| \frac{1}{D_{\mathbf{G}} \bar{D}_{\mathcal{M}}} \sum_{j=1}^{N} \left[ (\bar{D}_{\mathcal{M}} \gamma_{ij} - \bar{D}_{\mathcal{M}} \tilde{\gamma}_{ij}) + (\bar{D}_{\mathcal{M}} \tilde{\gamma}_{ij} - D_{\mathbf{G}} \tilde{\gamma}_{ij}) \right] \mathbf{V} f(x_j) \right\| \qquad (103)$$

$$= \left\| \frac{1}{D_{\mathbf{G}} \bar{D}_{\mathcal{M}}} \sum_{j=1}^{N} \left[ \bar{D}_{\mathcal{M}} (\gamma_{ij} - \tilde{\gamma}_{ij}) (\bar{D}_{\mathcal{M}} - D_{\mathbf{G}}) \tilde{\gamma}_{ij} \right] \mathbf{V} f(x_j) \right\| \qquad (104)$$

$$\leq \sum_{j=1}^{N} \left\| \frac{\bar{D}_{\mathcal{M}}}{D_{\mathbf{G}} \bar{D}_{\mathcal{M}}} \left[ (\gamma_{ij} - \tilde{\gamma}_{ij}) \right] \mathbf{V} f(x_j) \right\| + \left\| \frac{\tilde{\gamma}_{ij} (\bar{D}_{\mathcal{M}} - D_{\mathbf{G}})}{D_{\mathbf{G}} \bar{D}_{\mathcal{M}}} \mathbf{V} f(x_j) \right\| \qquad (105)$$

$$\leq \sum_{j=1}^{N} \|\mathbf{V}\| \|f(x_j)\| \left[ \frac{1}{D_{\mathbf{G}}} |\gamma_{ij} - \tilde{\gamma}_{ij}| + \frac{\tilde{\gamma}_{ij}}{D_{\mathbf{G}} \bar{D}_{\mathcal{M}}} |\bar{D}_{\mathcal{M}} - D_{\mathbf{G}}| \right] \qquad (106)$$

$$\leq C_V B_f \sum_{j=1}^{N} \left[ \frac{1}{D_{\mathbf{G}}} |\gamma_{ij} - \tilde{\gamma}_{ij}| + \frac{\tilde{\gamma}_{ij}}{D_{\mathbf{G}} \bar{D}_{\mathcal{M}}} |\bar{D}_{\mathcal{M}} - D_{\mathbf{G}}| \right] \qquad (107)$$

$$\leq C_V B_f \left[ \frac{N \cdot 2 e^M C_{QK} \Delta_{\mathrm{PE}}}{D_{\mathbf{G}}} + \frac{|\bar{D}_{\mathcal{M}} - D_{\mathbf{G}}|}{D_{\mathbf{G}} \bar{D}_{\mathcal{M}}} \sum_{j=1}^{N} \tilde{\gamma}_{ij} \right] \qquad (108)$$

$$\leq C_V B_f \left[ 2 e^{2M} C_{QK} \Delta_{\mathrm{PE}} + \frac{|\bar{D}_{\mathcal{M}} - D_{\mathbf{G}}|}{D_{\mathbf{G}}} \right] \qquad (109)$$

On (104) we rearrange the terms, on (105) we apply the triangle inequality, on (106) we use the operator norm inequality, and on (107) we apply the bounds from Assumptions 2 and **??**. On (108) we apply the bound from Lemma 7. Finally, (109) uses $\sum_{j=1}^{N} \tilde{\gamma}_{ij} = \bar{D}_{\mathcal{M}}$ for the second term, and for the first term we note that the worst case denominator is $D_{\mathbf{G}} \geq N \cdot e^{-M}$, which we can rearrange to bound $N/D_{\mathbf{G}} \leq e^M$.

We can bound the remaining term using Lemma 7:

$$\frac{|\bar{D}_{\mathcal{M}} - D_{\mathbf{G}}|}{D_{\mathbf{G}}} = \frac{1}{D_{\mathbf{G}}} \left| \sum_{j=1}^{N} \tilde{\gamma}_{ij} - \sum_{j=1}^{N} \gamma_{ij} \right| \qquad (110)$$

$$\leq \frac{1}{D_{\mathbf{G}}} \sum_{j=1}^{N} |\tilde{\gamma}_{ij} - \gamma_{ij}| \qquad (111)$$

$$\leq \frac{N}{D_{\mathbf{G}}} \cdot 2 e^M C_{QK} \Delta_{\mathrm{PE}} \qquad (112)$$

$$\leq 2 e^{2M} C_{QK} \Delta_{\mathrm{PE}}, \qquad (113)$$

where in (111) we apply the triangle inequality, in (112) we apply Lemma 7, and in (113) we again use $N/D_{\mathbf{G}} \leq e^M$.

Substituting (113) into (109), the second term of (101) is bounded by

$$\left\| \frac{1}{D_{\mathbf{G}}\bar{D}_{\mathcal{M}}} \sum_{j=1}^{N} (\bar{D}_{\mathcal{M}}\gamma_{ij} - D_{\mathbf{G}}\tilde{\gamma}_{ij})\mathbf{V}f(x_j) \right\| \le C_V B_f \left[ 2e^{2M}C_{QK}\Delta_{\mathrm{PE}} + 2e^{2M}C_{QK}\Delta_{\mathrm{PE}} \right] \tag{114}$$

$$= 4B_f e^{2M} C_{QKV}\Delta_{\mathrm{PE}}, \tag{115}$$

where we absorb the linear operator constants into $C_{QKV} = C_{QK}C_V$. Combining (102) and (115), we conclude the bound for the first term of (97) is

$$\left| \mathbf{\Phi}_{\mathbf{G}}(\mathbf{X};\mathbf{T})(x) - \bar{\mathbf{\Phi}}_{\mathcal{M}}(\mathbf{X};\mathbf{T})(x) \right| \le C_V \Delta_{\mathrm{PE}} + 4B_f e^{2M} C_{QKV}\Delta_{\mathrm{PE}}. \tag{116}$$

We now bound the second term of (97). Let $\tilde{D}_{\mathcal{M}} = \sum_{j=1}^{n} \int_{V_j} \tilde{\gamma}_{ij} d\mu(y)$ denote the discrete approximation of $D_{\mathcal{M}}$ in integral form,

$$\|\bar{\mathbf{\Phi}}_{\mathcal{M}}(\mathbf{X};\mathbf{T})(x_i) - \mathbf{\Phi}_{\mathcal{M}}(f,\mathcal{L};\mathbf{T})(x_i)\| \tag{117}$$

$$= \left\| \frac{1}{\tilde{D}_{\mathcal{M}}} \sum_{j=1}^{n} \int_{V_j} \tilde{\gamma}_{ij}\mathbf{V}f(x_j)d\mu(y) - \frac{1}{D_{\mathcal{M}}} \int_{\mathcal{M}} \tilde{\gamma}_{iy}\mathbf{V}f(y)d\mu(y) \right\| \tag{118}$$

$$= \left\| \frac{1}{\tilde{D}_{\mathcal{M}}D_{\mathcal{M}}} \left( D_{\mathcal{M}} \sum_{j=1}^{n} \int_{V_j} \tilde{\gamma}_{ij}\mathbf{V}f(x_j)d\mu(y) - \tilde{D}_{\mathcal{M}} \int_{\mathcal{M}} \tilde{\gamma}_{iy}\mathbf{V}f(y)d\mu(y) \right) \right\| \tag{119}$$

$$= \left\| \frac{1}{\tilde{D}_{\mathcal{M}}D_{\mathcal{M}}} \left( D_{\mathcal{M}} \sum_{j=1}^{n} \int_{V_j} \tilde{\gamma}_{ij}\mathbf{V}f(x_j)d\mu(y) - \tilde{D}_{\mathcal{M}} \sum_{j=1}^{n} \int_{V_j} \tilde{\gamma}_{iy}\mathbf{V}f(y)d\mu(y) \right) \right\| \tag{120}$$

$$= \left\| \frac{1}{\tilde{D}_{\mathcal{M}}D_{\mathcal{M}}} \left( D_{\mathcal{M}} \sum_{j=1}^{n} \int_{V_j} \tilde{\gamma}_{ij}\mathbf{V}f(x_j)d\mu(y) - \tilde{D}_{\mathcal{M}}\tilde{\gamma}_{iy}\mathbf{V}f(y)d\mu(y) \right) \right\| \tag{121}$$

$$\le \frac{1}{\tilde{D}_{\mathcal{M}}D_{\mathcal{M}}} \sum_{j=1}^{n} \int_{V_j} \left\| D_{\mathcal{M}}\tilde{\gamma}_{ij}\mathbf{V}f(x_j) - \tilde{D}_{\mathcal{M}}\tilde{\gamma}_{iy}\mathbf{V}f(y) \right\| d\mu(y) \tag{122}$$

In (119) we distribute the denominators, in (120) we partition the integral, in (121) we rearrange terms, and in (122) we apply the triangle inequality.

To bound the integrand of (122), we add and subtract $D_{\mathcal{M}}\tilde{\gamma}_{ij}\mathbf{V}f(y)$ and apply the triangle inequality:

$$\left\| D_{\mathcal{M}}\tilde{\gamma}_{ij}\mathbf{V}f(x_j) - \tilde{D}_{\mathcal{M}}\tilde{\gamma}_{iy}\mathbf{V}f(y) \right\| \tag{123}$$

$$\le \| D_{\mathcal{M}}\tilde{\gamma}_{ij}\mathbf{V}f(x_j) - D_{\mathcal{M}}\tilde{\gamma}_{ij}\mathbf{V}f(y) \| + \left\| D_{\mathcal{M}}\tilde{\gamma}_{ij}\mathbf{V}f(y) - \tilde{D}_{\mathcal{M}}\tilde{\gamma}_{iy}\mathbf{V}f(y) \right\| \tag{124}$$

$$\le D_{\mathcal{M}} \| \mathbf{V}(f(x_j) - f(y)) \| + \left\| (D_{\mathcal{M}}\tilde{\gamma}_{ij} - \tilde{D}_{\mathcal{M}}\tilde{\gamma}_{iy})\mathbf{V}f(y) \right\|. \tag{125}$$

The first term of (125) is bounded by

$$D_{\mathcal{M}} \| \mathbf{V}(f(x_j) - f(y)) \| \le D_{\mathcal{M}} \|\mathbf{V}\| \| f(x_j) - f(y) \| \tag{126}$$

$$\le D_{\mathcal{M}} C_V \| f(x_j) - f(y) \| \tag{127}$$

$$\le D_{\mathcal{M}} C_V r. \tag{128}$$

(126) linear operator bound, (127) Assumption 2 (128) using the fact that $\| f(x_j) - f(y) \| \le r$ for all $y \in V_j$, and Assumption 1 ($f$ is normalized Lipschitz continuous).

The second term of (125) is bounded by

$$\left\| (D_{\mathcal{M}}\tilde{\gamma}_{ij} - \tilde{D}_{\mathcal{M}}\tilde{\gamma}_{iy})\mathbf{V}f(y) \right\| \leq \left| D_{\mathcal{M}}\tilde{\gamma}_{ij} - \tilde{D}_{\mathcal{M}}\tilde{\gamma}_{iy} \right| \|\mathbf{V}\| \|f(y)\| \tag{129}$$

$$\leq C_V B_f \left| D_{\mathcal{M}}\tilde{\gamma}_{ij} - \tilde{D}_{\mathcal{M}}\tilde{\gamma}_{iy} \right| \tag{130}$$

$$= C_V B_f \left| D_{\mathcal{M}}\tilde{\gamma}_{ij} - D_{\mathcal{M}}\tilde{\gamma}_{iy} + D_{\mathcal{M}}\tilde{\gamma}_{iy} - \tilde{D}_{\mathcal{M}}\tilde{\gamma}_{iy} \right| \tag{131}$$

$$\leq C_V B_f \left( D_{\mathcal{M}} |\tilde{\gamma}_{ij} - \tilde{\gamma}_{iy}| + \left| D_{\mathcal{M}} - \tilde{D}_{\mathcal{M}} \right| \tilde{\gamma}_{iy} \right) \tag{132}$$

In (129) we apply the linear operator bound, in (130) we use Assumption 2 ($\|\mathbf{V}\| \leq C_V$) and $\|f(y)\| \leq B_f$, in (131) we add and subtract $D_{\mathcal{M}}\tilde{\gamma}_{iy}$, and in (132) we apply the triangle inequality and factor.

To bound the difference between denominator terms:

$$\left| D_{\mathcal{M}} - \tilde{D}_{\mathcal{M}} \right| = \left| \int_{\mathcal{M}} \tilde{\gamma}_{iy} d\mu(y) - \sum_{j=1}^{n} \int_{V_j} \tilde{\gamma}_{ij} d\mu(y) \right| \tag{133}$$

$$= \left| \sum_{j=1}^{n} \int_{V_j} (\tilde{\gamma}_{iy} - \tilde{\gamma}_{ij}) d\mu(y) \right| \tag{134}$$

$$\leq \sum_{j=1}^{n} \int_{V_j} |\tilde{\gamma}_{iy} - \tilde{\gamma}_{ij}| \, d\mu(y) \tag{135}$$

$$\leq \sum_{j=1}^{n} \int_{V_j} e^M B_f C_{QK} r \, d\mu(y) \tag{136}$$

$$= e^M B_f C_{QK} r, \tag{137}$$

where in (134) we partition the integral, in (135) we apply the triangle inequality, and in (136) we apply Lemma 6 and the fact that $\mu(\mathcal{M}) = 1$.

Now, applying (137) and Lemma (6) to (132) we have that the second term of (125) is bounded by

$$\left\| (D_{\mathcal{M}}\tilde{\gamma}_{ij} - \tilde{D}_{\mathcal{M}}\tilde{\gamma}_{iy})\mathbf{V}f(y) \right\| \leq C_V B_f \left( D_{\mathcal{M}} e^M B_f C_Q C_K \, r + e^M B_f C_{QK} r \tilde{\gamma}_{iy} \right) \tag{138}$$

$$\leq D_{\mathcal{M}} e^M B_f^2 C_{QKV} r + e^M B_f^2 C_{QKV} r \tilde{\gamma}_{iy} \tag{139}$$

Now, we return to bound the integral of (122),

$$\| \bar{\boldsymbol{\Phi}}_{\mathcal{M}}(\mathbf{X}; \mathbf{T})(x_i) - \boldsymbol{\Phi}_{\mathcal{M}}(f, \mathcal{L}; \mathbf{T})(x_i) \| \tag{140}$$

$$\leq \frac{1}{\tilde{D}_{\mathcal{M}} D_{\mathcal{M}}} \sum_{j=1}^{n} \int_{V_j} \left\| D_{\mathcal{M}}\tilde{\gamma}_{ij}\mathbf{V}f(x_j) - \tilde{D}_{\mathcal{M}}\tilde{\gamma}_{iy}\mathbf{V}f(y) \right\| d\mu(y) \tag{141}$$

$$\leq \frac{1}{\tilde{D}_{\mathcal{M}} D_{\mathcal{M}}} \sum_{j=1}^{n} \int_{V_j} \left[ D_{\mathcal{M}} C_V r + D_{\mathcal{M}} e^M B_f^2 C_{QKV} r + e^M B_f^2 C_{QKV} r \tilde{\gamma}_{iy} \right] d\mu(y) \tag{142}$$

$$\leq \frac{1}{\tilde{D}_{\mathcal{M}} D_{\mathcal{M}}} \left[ D_{\mathcal{M}} C_V r \mu(\mathcal{M}) + D_{\mathcal{M}} e^M B_f^2 C_{QKV} r \mu(\mathcal{M}) + e^M B_f^2 C_{QKV} r \sum_{j=1}^{n} \int_{V_j} \tilde{\gamma}_{iy} d\mu(y) \right] \tag{143}$$

$$= \frac{C_V r \mu(\mathcal{M})}{\tilde{D}_{\mathcal{M}}} + \frac{e^M B_f^2 C_{QKV} r \mu(\mathcal{M})}{\tilde{D}_{\mathcal{M}}} + \frac{e^M B_f^2 C_{QKV} r \mu(\mathcal{M})}{D_{\mathcal{M}}} \tag{144}$$

$$\leq \frac{C_V r \mu(\mathcal{M})}{e^{-M} \mu(\mathcal{M})} + \frac{e^M B_f^2 C_{QKV} r \mu(\mathcal{M})}{e^{-M} \mu(\mathcal{M})} + \frac{e^M B_f^2 C_{QKV} r \mu(\mathcal{M})}{e^{-M} \mu(\mathcal{M})} \tag{145}$$

$$\leq e^M C_V r + 2e^{2M} B_f^2 C_{QKV} r \tag{146}$$

where in (142) we use the bounds of (128) and (139), in (143) we evaluate the integrals of the first to terms, in (145) we distribute the denominators and use the fact that both denominators are positive quantities greater than $e^{-M}\mu(\mathcal{M})$.

We conclude that the second term of (97) is bounded by

$$\|\bar{\boldsymbol{\Phi}}_{\mathcal{M}}(\mathbf{X};\mathbf{T})(x_i) - \boldsymbol{\Phi}_{\mathcal{M}}(f,\mathcal{L};\mathbf{T})(x_i)\| \leq e^M C_V r + 2e^{2M} B_f^2 C_{QKV} r \tag{147}$$

Finally, we can bound (97) using (116) and (147), which by gathering terms and setting $B_f = 1$. With this we conclude that the output difference of GT and MT is bounded by

$$\|\boldsymbol{\Phi}_G(\mathbf{X};\mathbf{T})(x_i) - \boldsymbol{\Phi}_{\mathcal{M}}(f;\mathbf{T})(x_i)\| \leq C_V \Delta_{\text{PE}} + 4e^{2M} B_f C_{QKV} \Delta_{\text{PE}} + e^M C_V r + 2e^{2M} B_f^2 C_{QKV} r \tag{148}$$

$$\leq \left[ C_V + 4e^{2M} C_{QKV} \right] \Delta_{\text{PE}} + \left[ e^M C_V + 2e^{2M} C_{QKV} \right] r. \tag{149}$$

which gives us the statement of Theorem 1. $\qquad\square$

## 7. Proof of Corollary 2

Corollary 2 bounds the output difference between two GT's trained with differently sized graphs by applying Theorem 1.

*Proof.* The output difference can be bounded as

$$\frac{1}{\mu(\mathcal{M})} \|\mathbf{I}_{N_1} \boldsymbol{\Phi}_{\mathbf{G}_1}(\mathbf{X}_1;\mathbf{T}) - \mathbf{I}_{N_2} \boldsymbol{\Phi}_{\mathbf{G}_2}(\mathbf{X}_2;\mathbf{T})\|_{L^{1,2}(\mathcal{M})}$$

$$= \frac{1}{\mu(\mathcal{M})} \|\mathbf{I}_{N_1} \boldsymbol{\Phi}_{\mathbf{G}_1}(\mathbf{X}_1;\mathbf{T}) - \boldsymbol{\Phi}_{\mathcal{M}}(f;\mathbf{T}) + \boldsymbol{\Phi}_{\mathcal{M}}(f;\mathbf{T}) - \mathbf{I}_{N_2} \boldsymbol{\Phi}_{\mathbf{G}_2}(\mathbf{X}_2;\mathbf{T})\|_{L^{1,2}(\mathcal{M})} \tag{150}$$

$$\leq \frac{1}{\mu(\mathcal{M})} \|\mathbf{I}_{N_1} \boldsymbol{\Phi}_{\mathbf{G}_1}(\mathbf{X}_1;\mathbf{T}) - \boldsymbol{\Phi}_{\mathcal{M}}(f;\mathbf{T})\|_{L^{1,2}(\mathcal{M})} + \frac{1}{\mu(\mathcal{M})} \|\boldsymbol{\Phi}_{\mathcal{M}}(f;\mathbf{T}) - \mathbf{I}_{N_2} \boldsymbol{\Phi}_{\mathbf{G}_2}(\mathbf{X}_2;\mathbf{T})\|_{L^{1,2}(\mathcal{M})} \tag{151}$$

$$\leq \frac{1}{\mu(\mathcal{M})} \int_{\mathcal{M}} \|\mathbf{I}_{N_1} \boldsymbol{\Phi}_{\mathbf{G}_1}(\mathbf{X}_1;\mathbf{T})(x) - \boldsymbol{\Phi}_{\mathcal{M}}(f;\mathbf{T})(x)\|_2 \, d\mu(x) \tag{152}$$

$$+ \frac{1}{\mu(\mathcal{M})} \int_{\mathcal{M}} \|\boldsymbol{\Phi}_{\mathcal{M}}(f;\mathbf{T})(x) - \mathbf{I}_{N_2} \boldsymbol{\Phi}_{\mathbf{G}_2}(\mathbf{X}_2;\mathbf{T})(x)\|_2 \, d\mu(x) \tag{153}$$

where in (150) we add and subtract $\boldsymbol{\Phi}_{\mathcal{M}}(f;,\mathbf{T})$, in (151) we apply the triangle inequality, and (152) use the definition of $L^{1,2}(\mathcal{M})$.

The two terms in (153) correspond to the pointwise difference between the induced manifold signal of GT and the output signal of the MT, for $x \in \mathcal{M}$. Consider bounding the first term, that compares $\boldsymbol{\Phi}_{\mathbf{G}_1}$ with $\boldsymbol{\Phi}_{\mathcal{M}}$,

$$\frac{1}{\mu(\mathcal{M})} \int_{\mathcal{M}} \|\mathbf{I}_{N_1} \boldsymbol{\Phi}_{\mathbf{G}_1}(\mathbf{X}_1;\mathbf{T})(x) - \boldsymbol{\Phi}_{\mathcal{M}}(f;\mathbf{T})(x)\|_2 \, d\mu(x)$$

$$= \frac{1}{\mu(\mathcal{M})} \sum_{i=1}^{N} \int_{V_i} \|\boldsymbol{\Phi}_{\mathbf{G}_1}(\mathbf{X}_1;\mathbf{T})(x_i) - \boldsymbol{\Phi}_{\mathcal{M}}(f;\mathbf{T})(x)\|_2 \, d\mu(x) \tag{154}$$

$$= \frac{1}{\mu(\mathcal{M})} \sum_{i=1}^{N} \int_{V_i} \|\boldsymbol{\Phi}_{\mathbf{G}_1}(\mathbf{X}_1;\mathbf{T})(x_i) - \boldsymbol{\Phi}_{\mathcal{M}}(f;\mathbf{T})(x_i) + \boldsymbol{\Phi}_{\mathcal{M}}(f;\mathbf{T})(x_i) - \boldsymbol{\Phi}_{\mathcal{M}}(f;\mathbf{T})(x)\|_2 \, d\mu(x) \tag{155}$$

$$\leq \frac{1}{\mu(\mathcal{M})} \sum_{i=1}^{N} \int_{V_i} \|\boldsymbol{\Phi}_{\mathbf{G}_1}(\mathbf{X}_1;\mathbf{T})(x_i) - \boldsymbol{\Phi}_{\mathcal{M}}(f;\mathbf{T})(x_i)\| + \|\boldsymbol{\Phi}_{\mathcal{M}}(f;\mathbf{T})(x_i) - \boldsymbol{\Phi}_{\mathcal{M}}(f;\mathbf{T})(x)\|_2 \, d\mu(x) \tag{156}$$

In (154) we apply the interpolator operator on $\boldsymbol{\Phi}_{\mathbf{G}_1}$, then in (156) we add and subtract the MT output at point $x_i$, and apply triangular inequality. The first term of (156) is the statement of Theorem 1. The second term is the output difference

of MT between a point within a partition $x \in V_i$ and the center of the ball containing the partition $x_i$, therefore it holds that $\|x_i - x\| \le r$. Denote the softmax denominators over $x$ and $x_i$ as $D_{\mathcal{M}}(x)$ and $D_{\mathcal{M}}(x_i)$ respectively. We can decompose this as

$$\|\mathbf{\Phi}_{\mathcal{M}}(f; \mathbf{T})(x_i) - \mathbf{\Phi}_{\mathcal{M}}(f; \mathbf{T})(x)\|_2$$

$$\le \left\| \frac{\int_{\mathcal{M}} \tilde{\gamma}_{iy} \mathbf{V} f(y) d\mu(y)}{\int_{\mathcal{M}} \tilde{\gamma}_{iy} d\mu(y)} - \frac{\int_{\mathcal{M}} \tilde{\gamma}_{xy} \mathbf{V} f(y) d\mu(y)}{\int_{\mathcal{M}} \tilde{\gamma}_{xy} d\mu(y)} \right\| \tag{157}$$

$$\le \left\| \frac{1}{D_{\mathcal{M}}(x) D_{\mathcal{M}}(x_i)} \int_{\mathcal{M}} \left( D_{\mathcal{M}}(x) \tilde{\gamma}_{iy} \mathbf{V} f(y) - D_{\mathcal{M}}(x_i) \tilde{\gamma}_{xy} \mathbf{V} f(y) \right) d\mu(y) \right\| \tag{158}$$

$$\le \frac{1}{D_{\mathcal{M}}(x) D_{\mathcal{M}}(x_i)} \int_{\mathcal{M}} \left\| D_{\mathcal{M}}(x) \tilde{\gamma}_{iy} \mathbf{V} f(y) - D_{\mathcal{M}}(x_i) \tilde{\gamma}_{xy} \mathbf{V} f(y) \right\| d\mu(y) \tag{159}$$

Which we obtain by distributing the denominators, applying the triangle inequality, and grouping terms. The integrand in (159) is bounded as

$$\|D_{\mathcal{M}}(x) \tilde{\gamma}_{iy} \mathbf{V} f(y) - D_{\mathcal{M}}(x_i) \tilde{\gamma}_{xy} \mathbf{V} f(y)\|$$

$$\le \|D_{\mathcal{M}}(x) \tilde{\gamma}_{iy} \left[ \mathbf{V} f(x) - \mathbf{V} f(y) \right] - \left[ D_{\mathcal{M}}(x) \tilde{\gamma}_{iy} - D_{\mathcal{M}}(x_i) \tilde{\gamma}_{xy} \right] \mathbf{V} f(y)\| \tag{160}$$

$$\le D_{\mathcal{M}}(x) C_V \tilde{\gamma}_{iy} \|f(x) - f(y)\| - C_V B_f \left| D_{\mathcal{M}}(x) \tilde{\gamma}_{iy} - D_{\mathcal{M}}(x_i) \tilde{\gamma}_{xy} \right| \tag{161}$$

where in (160) add and subtract subtract $D_{\mathcal{M}}(x) \tilde{\gamma}_{iy} \mathbf{V} f(x)$

We now focus on the second term of (161),

$$\|D_{\mathcal{M}}(x) \tilde{\gamma}_{iy} - D_{\mathcal{M}}(x_i) \tilde{\gamma}_{xy}\| \le D_{\mathcal{M}}(x) |\tilde{\gamma}_{iy} - \tilde{\gamma}_{xy}| + \| \left[ D_{\mathcal{M}}(x) - D_{\mathcal{M}}(x_i) \right] \tilde{\gamma}_{xy} \| \tag{162}$$

The second term of (162) is

$$\tilde{\gamma}_{xy} \|D_{\mathcal{M}}(x) - D_{\mathcal{M}}(x_i)\| = \left\| \int_{\mathcal{M}} \tilde{\gamma}_{xy} \tilde{\gamma}_{xy} - \tilde{\gamma}_{iy} d\mu(y) \right\| \tag{163}$$

$$\le \int_{\mathcal{M}} \tilde{\gamma}_{xy} \|\tilde{\gamma}_{xy} - \tilde{\gamma}_{iy}\| d\mu(y) \tag{164}$$

$$\le D_{\mathcal{M}}(x_i) e^M C_{QK} B_f r \tag{165}$$

where we use Lemma 8 to bound the remaining integral by $D_{\mathcal{M}}(x_i)$.

Returning to the integral (159), we have

$$\frac{C_V B_f}{D_{\mathcal{M}}(x) D_{\mathcal{M}}(x_i)} \int_{\mathcal{M}} \left\| D_{\mathcal{M}}(x) \tilde{\gamma}_{iy} \mathbf{V} f(y) - D_{\mathcal{M}}(x_i) \tilde{\gamma}_{xy} \mathbf{V} f(y) \right\| d\mu(y) \tag{166}$$

$$\le \frac{C_V B_f}{D_{\mathcal{M}}(x) D_{\mathcal{M}}(x_i)} \int_{\mathcal{M}} \left\| D_{\mathcal{M}}(x) e^M C_{QK} B_f r + \tilde{\gamma}_{xy} \mu(\mathcal{M}) e^M C_{QK} B_f r \right\| d\mu(y) \tag{167}$$

$$= \frac{C_V B_f}{D_{\mathcal{M}}(x) D_{\mathcal{M}}(x_i)} D_{\mathcal{M}}(x) \mu(\mathcal{M}) e^M C_{QK} B_f r + \tilde{\gamma}_{xy} D_{\mathcal{M}}(x) \mu(\mathcal{M}) e^M C_{QK} B_f r \tag{168}$$

$$= \frac{1}{D_{\mathcal{M}}(x_i)} 2 \mu(\mathcal{M}) e^M C_{QKV} B_f^2 r \tag{169}$$

$$\le 2 e^{2M} C_{QKV} B_f^2 r \tag{170}$$

by applying the bounds of (165) and Lemma 8, then evaluating the integral, and finally using the fact that the denominator is $D_{\mathcal{M}}(x_i) \ge e^M / \mu(\mathcal{M})$. We conclude that

$$\|\mathbf{\Phi}_{\mathcal{M}}(f; \mathbf{T})(x_i) - \mathbf{\Phi}_{\mathcal{M}}(f; \mathbf{T})(x)\|_2 \le 2 e^{2M} C_{QKV} B_f^2 r \tag{171}$$

We can now bound (152) by applying this bound and the bound of Theorem 1,

$$\frac{1}{\mu(\mathcal{M})} \int_{\mathcal{M}} \|\mathbf{I}_{N_1} \mathbf{\Phi}_{\mathbf{G}_1}(\mathbf{X}_1; \mathbf{T})(x) - \mathbf{\Phi}_{\mathcal{M}}(f; \mathbf{T})(x)\|_2 \, d\mu(x) \leq \Delta_{\mathbf{\Phi}(\mathbf{X}_1)} + 2e^{2M} C_{QKV} B_f^2 r, \tag{172}$$

where $\Delta_{\mathbf{\Phi}(\mathbf{X}_1)}$ denotes the bound of Theorem 1 for $G_1$.

Applying this bound in (153) for $\mathbf{\Phi}(\mathbf{X}_1)$ and $\mathbf{\Phi}(\mathbf{X}_2)$ gives us the statement of Corollary 2.

$\square$

### 7.1. Proof of Corollary 4

*Proof.* The proof of Theorem 1 can be adapted to account for Sparse GT. Notice that with the definition of the $k$-hop neighborhood, the only terms contributing to the softmax are $|\mathcal{N}^{\leq k}(x_i)|$. Similarly for the restricted MT, the only nonzero terms contributing to t he integral are those contained in partitions $y \in V_i \subset B_r(x_i)$, i.e. $D_{\mathcal{M}} = \int_{V_i} \tilde{\gamma}_{iy} V f(y) d\mu(y)$. With those modifications, the same proof applies and we arrive at the same bound. $\square$

## 8. Experiment Implementation Details

**Datasets.** SNAP-Patents is a network for patents granted between 1963 to 1999 in the US. Each node is a patent, and a directed edge connects a patent to another patent that it cited. The prediction task is to classify each patent into one of five time intervals. ArXiv-year is a paper citation network on the arXiv papers. Each node represents a paper, and a directed edge connects a paper to another paper that it cited. The task is to predict the posting time of each paper, which is classified into one of five time intervals between 2013 and 2020. Both SNAP-Patents and arXiv-year are heterophilic graph datasets. OGBN-MAG is a heterogeneous network composed of a subset of the Microsoft Academic Graph (MAG), capturing the relationships among papers, authors, institutions and topics. The node classification task is to predict the venue of each paper. Reddit-Binary consists of 2000 graphs, each representing a subreddit community. The graph-level binary classification task is to identify each community as either a Q&A community or discussion-based community using graph structures.

*Table 2.* Datasets used for transferability experiments

| **Dataset** | Nodes | Edges | Max Train/Test Nodes | Classes | Graphs | Feature Dim |
|---|---|---|---|---|---|---|
| SNAP-Patents | 2,923,922 | 13,975,788 | 1,315,764 | 5 | 1 | 269 |
| ArXiv-year | 169,343 | 1,166,243 | 76,203 | 5 | 1 | 128 |
| OGBN-MAG | 736,389 | 5,416,271 | 2,923,922 | 349 | 1 | 128 |
| Reddit-Binary | avg. 429.61 | avg. 497.75 | / | 2 | 2000 | 0 |

**Dataset preparation.** Each dataset is split into train/val/test fractions of 45%-10%-45% respectively. For datasets that have pre-established train/test masks, we discard the masks in favor of our partition proportions. The training and testing partitions are the sources for the graph subsampling procedure explained in Section 4.

**Training procedure and transferability evaluation.** For each model architecture, we train multiple models with graphs of increasing sizes and evaluate each model on testing graphs of different sizes. For single-graph datasets, the training graphs and testing graphs are constructed by subsampling a fraction of nodes from the training split and the testing split respectively. For multi-graph datasets, we construct the training graphs and testing graphs by subsampling the nodes in each training graph and each testing graph respectively by a specific fraction. We do not create graph batches, but rather train with the full subsampled graph.

**Hyperparameters.** For each dataset, we tuned hyperparameters by training each model on the full training graph. Then, we held the selected hyperparameters fixed across all smaller training graph sizes. In general, We tuned learning rate over the interval $[1 \times 10^{-5}, 1 \times 10^{-2}]$, dropout and attention dropout over $[0, 0.5]$, the feedforward hidden dimension in $\{128, 256, 512\}$, the number of transformer heads in $\{4, 8\}$, the number of GNN/Transformer layers in $\{3, 4, 5, 6, 7\}$, the number of GNN/Sparse GT hops in $\{2, 3, 4\}$, and the Expander degree for Exphormer in $\{3, 4\}$. The full set of hyperparameters used for each model and dataset is available in Table 3.

*Table 3.* Hyperparameters for different model architectures across datasets

| Model/Hyperparameters | snap-patents | arXiv-year | OGBN-MAG | REDDIT-BINARY |
|---|---|---|---|---|
| **General** | | | | |
| Max epochs | 300 | 300 | 700 | 350 |
| **GNN** | | | | |
| Learning rate | $1 \times 10^{-3}$ | $1 \times 10^{-2}$ | $1 \times 10^{-2}$ | $3 \times 10^{-3}$ |
| Dropout | 0.25 | 0.5 | 0.25 | 0 |
| Hidden channels | 256 | 256 | 128 | 512 |
| Number of hop | 3 | 3 | 3 | 3 |
| Number of layers | 7 | 3 | 7 | 2 |
| **Sparse GT + RPEARL** | | | | |
| Learning rate | $5 \times 10^{-4}$ | $5 \times 10^{-4}$ | $5 \times 10^{-4}$ | $1 \times 10^{-5}$ |
| Dropout | 0.01 | 0.01 | 0.5 | 0.01 |
| Attention dropout | 0.01 | 0.01 | 0.05 | 0.01 |
| Dim model | 256 | 256 | 128 | 128 |
| Heads | 8 | 8 | 4 | 8 |
| Number of hops | 2 | 2 | 2 | 3 |
| Number of layers | 8 | 3 | 3 | 3 |
| **Dense GT** | | | | |
| Learning rate | $5 \times 10^{-4}$ | $5 \times 10^{-4}$ | $3 \times 10^{-4}$ | $5 \times 10^{-4}$ |
| Dropout | 0.05 | 0.05 | 0.05 | 0.1 |
| Attention dropout | 0.15 | 0.15 | 0.15 | 0.1 |
| Transformer Heads | 4 | 4 | 4 | 8 |
| Transformer dim feedforward | 128 | 128 | 128 | 512 |
| Transformer dim model | 256 | 256 | 128 | 128 |
| Transformer number of layers | 3 | 3 | 3 | 3 |
| Features | Data | Random | Data | Random |
| PE hidden channels | 128 | 128 | 128 | 512 |
| PE number of layers | 8 | 8 | 8 | 4 |
| **Exphormer** | | | | |
| Learning rate | $1 \times 10^{-3}$ | $1 \times 10^{-3}$ | $1 \times 10^{-3}$ | $5 \times 10^{-4}$ |
| Dropout | 0.5 | 0.5 | 0.5 | 0.01 |
| Dim model | 256 | 128 | 256 | 512 |
| Dim feedforward | 512 | 512 | 512 | 512 |
| Expander algorithm | Random-d | Random-d | Random-d | Random-d |
| Expander degree | 3 | 3 | 3 | 4 |
| Heads | 8 | 8 | 8 | 4 |
| Number of layers | 2 | 2 | 2 | 5 |
| **MLP** | | | | |
| Learning rate | $1 \times 10^{-3}$ | $1 \times 10^{-3}$ | $1 \times 10^{-3}$ | $5 \times 10^{-4}$ |
| Dropout | 0.5 | 0.5 | 0.5 | 0 |
| Hidden Dim | 256 | 256 | 128 | 512 |
| Number of layers | 3 | 3 | 5 | 3 |

## 8.1. Extended Results

In this section, we provide additional results, including transferability analysis for other reference architectures, as well as the full heatmaps for every model-dataset combination.

**Models.** In addition to DGT and Sparse GT, we consider MLP and GNN (Du et al., 2018) baselines, and Exphormer (Shirzad et al., 2023), a sparse variant of graph transformer that computes attention coefficients for (i) one-hop neighbors, (ii) random edges from an expander graph, (iii) $N$ additional attention coefficients between each node and a virtual global node. Crucially, Exphormer has no positional encodings beyond its masking.

**Transferability and comparison between models.** Figure 5 contains transferability plots for all architectures. We observe transferability patterns for Dense GT, Sparse GT and GNN in most datasets. Exphormer shows transferable behavior in MAG and SNAP. In all datasets, GT is either comparable (MAG,Reddit) or significantly superior (SNAP, Arxiv-year) to GNN. This performance gap is consistent with previous work's observations of the success of global attention in heterophilic datasets (Dwivedi & Bresson, 2020). The trend of Exphormer in ArXiV-year also shows a more pronounced monotonic increase, possibly due to the random sampling procedure being more beneficial on larger graphs. In MAG and Reddit, GNN and GTs show comparable performance and transferability patterns. As the training fraction increases, the total number of nodes decreases. In the cases of SNAP-patents and MAG, Exphormers presents a monotonically increasing performance as training graph size increases.

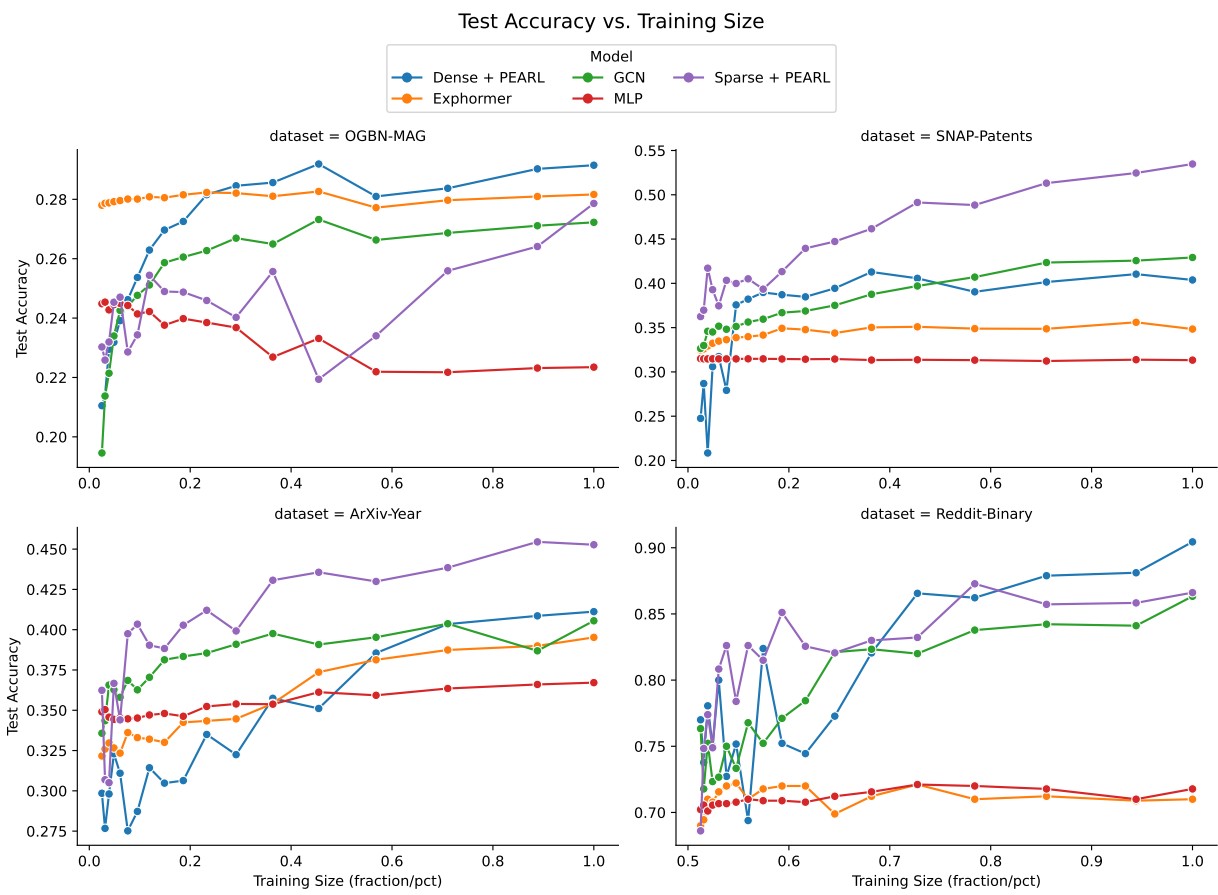

*Figure 5.* Transferability plots for all datasets and architectures. For each dataset, the $x$ axis represents the train graph sizes as a proportion of the largest graph ($\alpha$), and the $y$ axis is the test accuracy at the full-sized graph. The titles show dataset name and largest graph size.

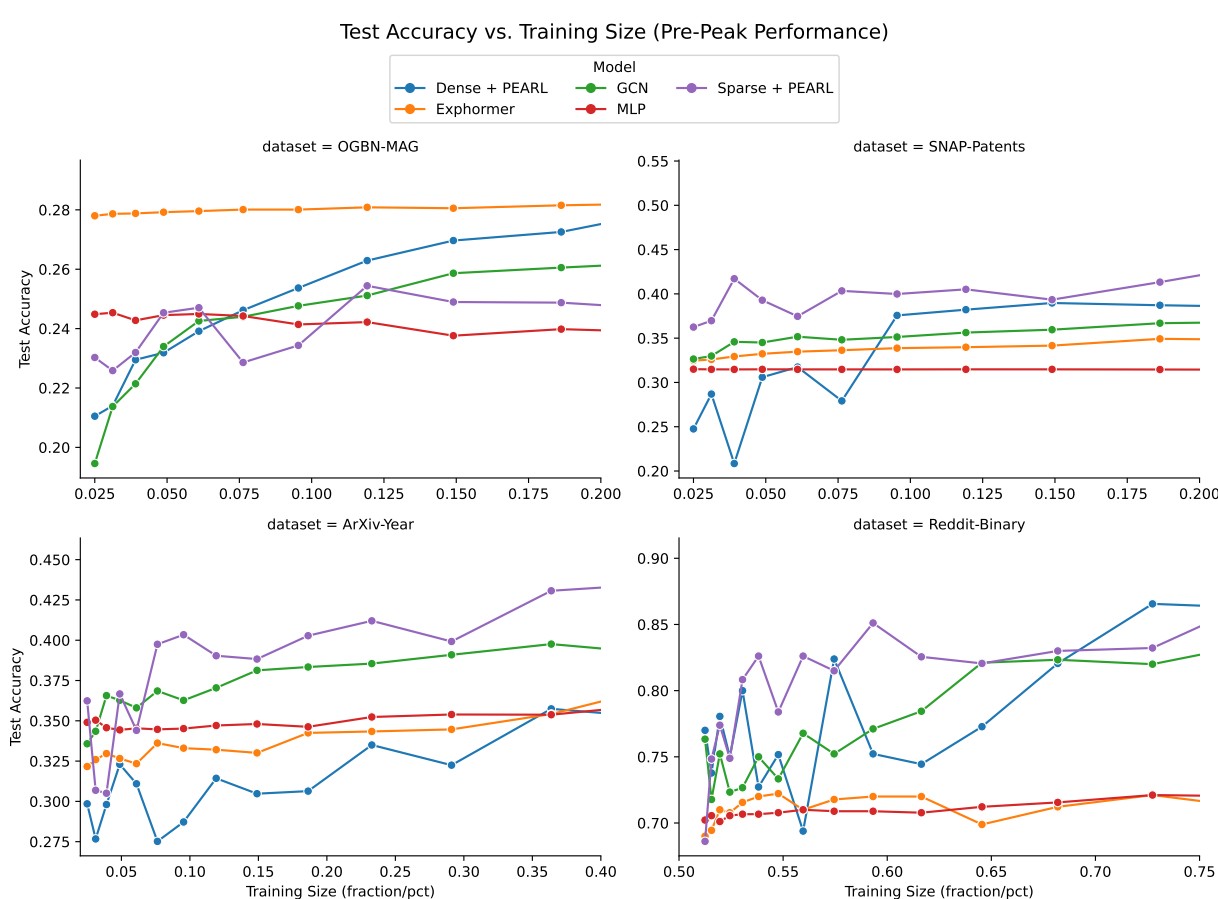

*Figure 6.* Transferability plots for all datasets and architectures, zoomed into pre-peak-performance training fractions. For each dataset, the $x$ axis represents the train graph sizes as a proportion of the largest graph ($\alpha$), and the $y$ axis is the test accuracy at the full-sized graph. The titles show dataset name and largest graph size.

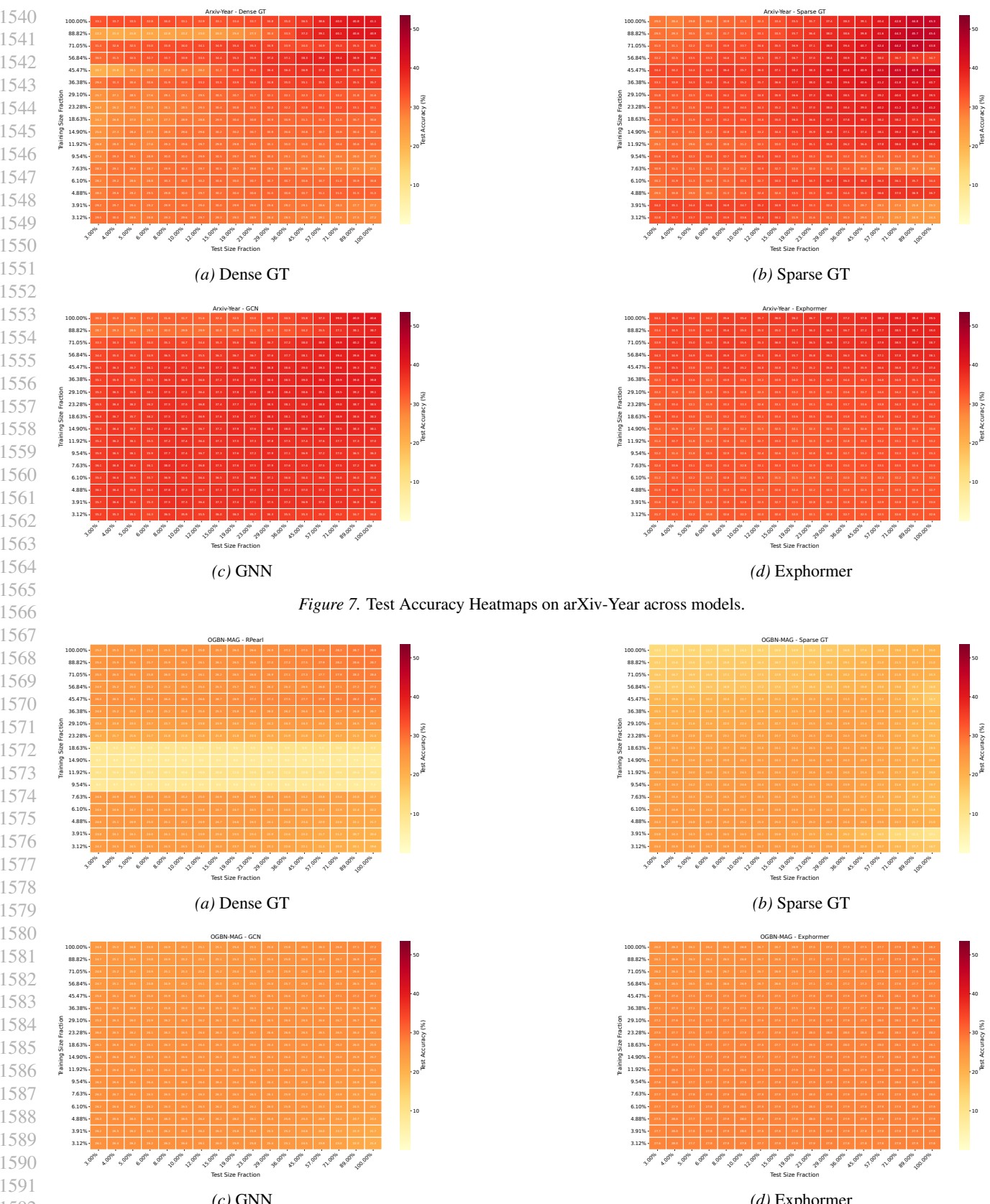

*(a)* Dense GT

*(b)* Sparse GT

*(c)* GNN

*(d)* Exphormer

*Figure 7.* Test Accuracy Heatmaps on arXiv-Year across models.

*(a)* Dense GT

*(b)* Sparse GT

*(c)* GNN

*(d)* Exphormer

*Figure 8.* Test Accuracy Heatmaps on OGBN-MAG across models.

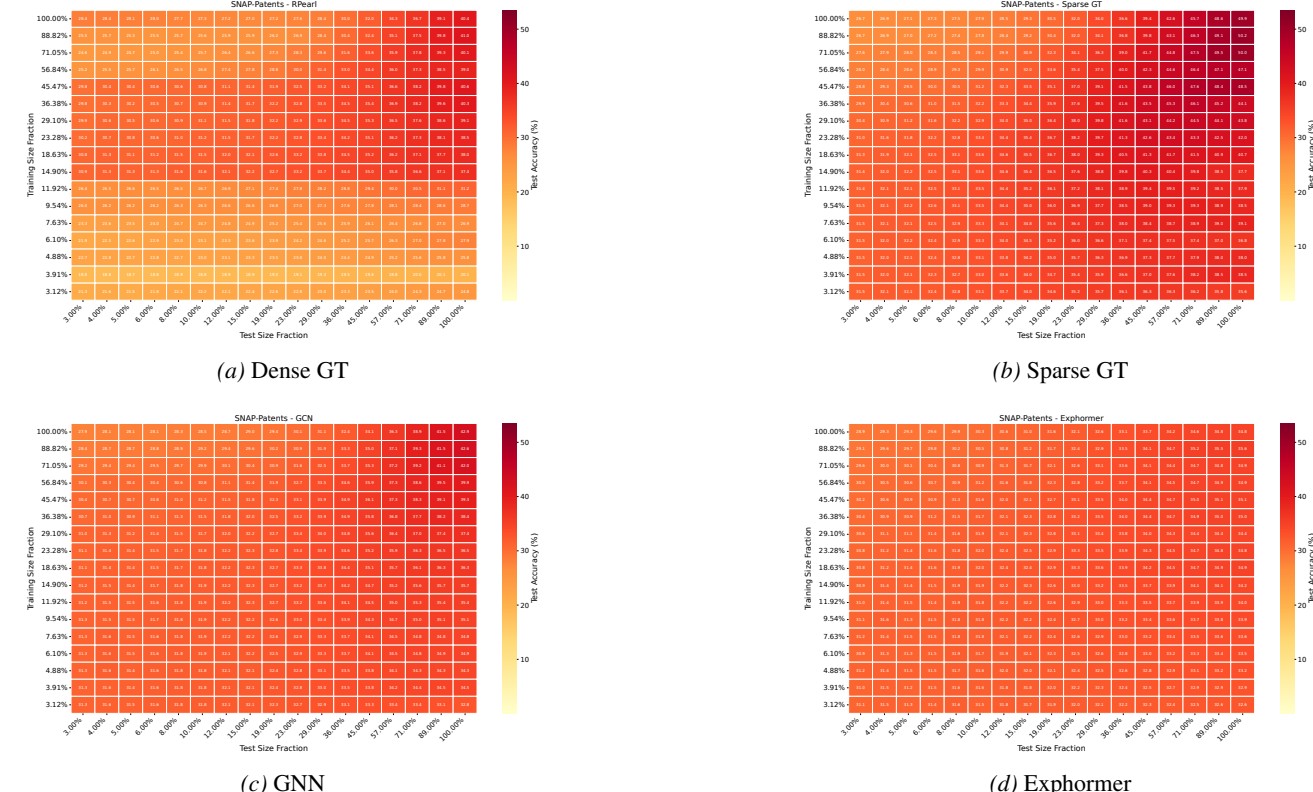

*(a)* Dense GT

*(b)* Sparse GT

*(c)* GNN

*(d)* Exphormer

*Figure 9.* Test Accuracy Heatmaps on SNAP-Patents across models.

*Table 4.* Runtimes in minutes for three subsample sizes of Sparse and Dense GT

| Dataset | Model | 6.10% | 56.84% | 100.00% |
|---------|-------|-------|--------|---------|
| ArXiv-Year | Dense GT | 51.6 | 10.0 | 19.8 |
| ArXiv-Year | Sparse GT | 8.8 | 1.4 | 2.0 |
| OGBN-MAG | Dense GT | 41.8 | 103.3 | 366.6 |
| OGBN-MAG | Sparse GT | 22.4 | 7.4 | 15.0 |
| SNAP-Patents | Dense GT | 223.3 | 945.7 | 2763.6 |
| SNAP-Patents | Sparse GT | 28.3 | 13.5 | 23.2 |

## 8.2. Runtime analysis

Our transferability results imply that it is possible to train with small graphs and attain competitive test performance on larger graphs, resulting in shorter training times. In addition to this, Sparse GT reduces the computational complexity from quadratic in the number of nodes to quadratic in the $k$-hop neighborhood cardinality. Table 4 shows the training times of Dense and Sparse GT for multiple fractions. We observe that Sparse GT is one or two orders of magnitude faster than Dense GT in all settings. It is worth highlighting that Sparse GT is both faster and achieves superior performance than Dense GT in ArXiv-year, and SNAP-Patents.

## 8.3. Terrain graphs

In Section 4.3 we presented a practical implication of our theoretical results using terrain graphs, which are approximations of terrain manifolds. Sparse GT provides an efficient, scalable, and expressive architecture that can be applied to the metric learning problem in terrain graphs. Furthermore, the transferability guarantees of Corollary 3 imply that we need not train with the highest possible resolution of terrain graph in order to achieve competitive performance. Here we provide implementation details and an additional result on a second terrain graph dataset.

**Metric learning.** The setup of (Chen et al., 2025) is to use imitation learning to learn a latent space that resembles true shortest-path distance between points. For two points $x_i, x_j$ in the terrain graph, we train to minimize $\|L_1(\phi(x_i) - \phi(x_j)) - SPD(x_i, x_j)\|_2$, where $\phi$ is an embedding function, in our case either GNN or SGT. On (Chen et al., 2025), a subsequent stage consists of freezing the embedding function's parameters and finetuning an additional MLP on a larger set of sample points. In our experiments, we omit this second stage.

**Datasets.** The terrain graphs we use are Digital Elevation Model (DEM) datasets. The results of Section 4.3 used a DEM of the Troms region of Norway (Kartverket (Norwegian Mapping Authority), 2025). The Norway dataset is available in a $2000 \times 2000$ (4M nodes) resolution. For our experiments we downsample so that the largest grid graph is $250 \times 250$.

**Graph and train/test data constructions.** To test transferability, we generate graphs of evenly spaced points based on the high resolution graph. The downsampling parameter $r$ controls the space between generated points, i.e. $r = 1$ is the full resolution, $r = 2$ takes every second point (resulting in a $1000 \times 1000$ graph, etc. The edges of the graph are taken to be the 8 nearest neighbors. For training datasets, we subsample 500 sources and 100 targets per source, for a total of $50,000$ pairs. For testing, a total of $10,000$ source points are selected based on maximum height, and the SPD is computed between these points and every other point in the graph.

