# OpenReview forum: "Size Transferability of Graph Transformers with Convolutional Positional Encodings"
_ICML.cc/2026/Conference — Submitted to ICML 2026_

### Official Review · Reviewer_7d7z · 2026-03-05

**Soundness:** 3
**Presentation:** 1
**Significance:** 2
**Originality:** 2
**Overall Recommendation:** 4
**Confidence:** 2

**Summary:**

The paper proposes a theoretical framework for understanding the transferability of graph transformers (GTs) from small graphs to larger graphs, using a manifold-based perspective. The authors show that graph transformers can inherit transferability when their positional encodings are generated by transferable GNNs, such as the RPEARL positional encoding.
 They provide theoretical guarantees that the approximation gap between GTs and MTs decays as the number of nodes in the graph grows, suggesting that training a GT on a small graph can be effectively transferred to larger graphs. The paper also provides empirical evaluations on several graph datasets, showing that graph transformers trained on smaller graphs achieve competitive performance when applied to larger graphs.

**Compliance With Llm Reviewing Policy:**

Affirmed.

**Final Justification:**

I thanks the authors for the clarification in the rebuttal. The paper provides interesting insight on the transferability of GTs, but with limited applications. For this reason I think the paper deserves a weak accept

**Key Questions For Authors:**

## 1. Intuition behind Theorem 1

The claim that the pointwise output difference between the graph transformer and the manifold transformer decreases as N increases is central to the paper. Could the authors provide additional intuition explaining why this convergence occurs and how it directly supports the claim of transferability from small to larger graphs?

## 2. Clarification of notation and missing definitions

Several quantities used in the theoretical analysis appear without clear definitions. Could the authors clarify:
- the definitions of H and Z in the positional encoding \Psi_G(H,L,Z),
- the precise definition of \Delta_{PE},
- the meaning of the probability parameter \delta used in the bounds?

## 3. Impact statement
Can the authors add the Impact statement section?

Providing clearer definitions would improve readability and understanding of the theoretical results.

**Limitations:**

The theoretical analysis in the paper focuses on a single type of positional encoding, namely the RPEARL positional encoding. While this choice is reasonable given its known transferability properties, it limits the generality of the proposed framework. It would be valuable to understand whether the theoretical results could be extended to other learnable positional encodings commonly used in graph learning.

For instance, approaches such as LSPE (Learnable Structural Positional Encoding) [Dwivedi et al., 2021] and GPSE (Graph Positional and Structural Encoder) [Cantürk et al., 2023] also learn structural or positional representations using neural architectures and are widely used in graph transformer models. Discussing whether the assumptions required by the theoretical framework hold for these methods, or whether the analysis could be generalized to a broader class of positional encodings, would strengthen the paper and increase the scope of its contributions.

**Strengths And Weaknesses:**

## Soundness:
The paper presents a theoretical framework for understanding the transferability of graph transformers, which is a valuable contribution. However, there are some concerns regarding the clarity of the theoretical results and the definitions of certain terms used in the theorems.
In particular, the claim in Line 250, Column 2 that “Theorem 1 indicates that the pointwise output difference of GT and MT decays as the size of the sampled graph N grows” is not entirely clear. Since this statement is central to the argument for the transferability of graph transformers, the paper would benefit from providing additional intuition or explanation of the mechanisms behind this result. In its current form, it is difficult to fully understand how the theorem supports the practical claim that models trained on small graphs can generalize to larger ones.

## Presentation:
The presentation could be improved. The paper contains several typos and unclear definitions that make parts of the theoretical analysis difficult to follow. In particular, some mathematical objects appear in the theorems without being clearly introduced.
 - Line 110, Column 2: H and Z are not defined
 - Line263, Column 2: is the delta_pe defined? What is it?
 - Line 266 Column 2 and 263 Column 2: what is the delta here?

These missing definitions make the theoretical exposition harder to follow. The narrative could be improved by ensuring that all variables and operators are formally introduced before they are used.

Additionally, there appear to be some inconsistencies in figure references:
- Line 367, Column 1: The figure reference is wrong, it should be Figure 2 and not Figure 4b
- Could be that the reference to FIgure7 on line 377 Column 1 is wrong and should be Figure 9?

Additionally there are some typos that should be corrected:
- Line 139: is the 1 inside the diag() a typo?
- Line 167, Column 2: typo "ove"
 - Line 356, Column 1: missing a ) after (Equation (1)

## Significance:
The paper provides theoretical understanding of the transferability of graph transformers with positional encodings generated by transferable GNNs. While the theoretical analysis may be of interest to researchers working on graph transformers, the scope of the contribution is relatively specialized within the broader graph learning community.

## Originality:
The paper claims to provide the first transferability analysis of graph transformers from a manifold perspective. However, the proposed theoretical framework builds heavily on existing work on graph-to-manifold convergence and transferability theory for GNNs. The main novelty lies in extending these ideas to graph transformers with GNN-based positional encodings rather than introducing a fundamentally new analytical framework. As a result, the degree of conceptual novelty appears somewhat limited.

---

> ### Author Rebuttal · Authors · 2026-03-31
>
> **Response to Soundness**
>
> The mechanism for transferability relies on Theorem 1, which states that the output difference between GT and MT vanishes as the graph size ($N$) grows. This means that the output difference between GT and its limit object (MT) become smaller with larger graphs. Consider two GTs trained on graphs with sizes $M$ and $N$ respectively, with $M<N$. Theorem 1 implies that, for $M$ and $N$ large enough, the output for both GTs will be close to the output of MT. Therefore, the output of the two GTs must necessarily also be close to each other. This is what allows us to prove Cor. 2, which states that the outputs of GTs on two different graph sizes are close because each is close to the same  limit. Since their output differences are small, their performance must also be similar. This matches what we verify empirically in our transferability experiments.
>
> We thank the reviewer for this question. We will revise the paper to make this interpretation more explicit.
>
> **Response to Presentation**
>
> We thank the reviewer for the thorough revision of our paper. We have addressed the typos and wrong Figure references. We would like to add some clarifications:
>  - $\mathbf{H}_k \in \mathrm{R}^{D \times 1}$ are the learnable matrices of the GNN,  defined in Line 142 Col 2, and $\mathbf{Z}$ is the input to the GNN, defined in Line 128 Col 2, and it is  either set to the node features $\mathbf{X}$ or sampled from an isotropic Gaussian distribution.
> - $\Delta_{\text{PE}}$ is a scalar quantity defined in Assumption 3 (Line 263). It is defined as the output difference between positional encoding $\psi_{\mathbf{G}}$ and its limit object $\psi_{\mathcal{M}}$. For instance for GNN-based PEs, it is the pointwise output difference between a GNN and a Manifold Neural Network (MNN), established in previous work. We are grateful to the reviewer for this observation, and we will add a clearer exposition of this fact to the Appendix.
> - $\Delta_{\mathbf{\Phi(X}}$ is the GT to MT convergence bound in Equation (10).
> - The $\mathbf{1}$ in $\text{diag}(\mathbf{A1})$ is the all ones vector of dimension $N$. This expression corresponds to the degree matrix.
>
> **Response to significance**
>
> We agree that our theory is centered around graph transformers with GNN encodings. However, we point out that it covers a wide range of GT backbones: e.g. GraphGPS, GT, Sparse GT, UnifiedGT, Exphormer and positional encodings (any variant that uses a GNN, e.g. GCN, GIN, GraphSAGE, etc. To emphasize the point, we focus on improving the computational efficiency of GTs with provable size-transferability, instead of proposing a better GT architecture.
>
> Furthermore, we have provided a set of additional experiments with a new backbone (GraphGPS), as well as discussion and theoretical results around other positional encodings (Laplacian PEs, Random Walk PEs, additional GNN-based PEs), which are of interest to a broad set of the graph ML community in general. See responses to PtBR (W1 - Other PEs, W2 - Other Backbones) for more details.
>
>
> **Response to Originality**
>
> The main novelty lies in extending these ideas to graph transformers with GNN-based positional encodings rather than introducing a fundamentally new analytical framework. As a result, the degree of conceptual novelty appears somewhat limited.
>
> Note that our theoretical results are entirely independent: the transferability of positional encodings is imported only as an assumption (Assumption 3). The key insight from our work is that graph transformers are transferable if this assumption on positional encodings is satisfied. Our conceptual novelty lies in showing that this theory extends nontrivially to graph transformers by introducing a manifold-transformer limit and proving an inheritance principle: if the positional encoding is transferable, then the full graph transformer is transferable. This is not immediate from prior GNN results because self-attention is a global, normalized nonlinear operator rather than a local convolution. We will therefore position the contribution more precisely as a manifold-based transferability analysis of graph transformers, with GNN-based positional encodings serving as a concrete analyzable instantiation.
>
>
> **Response to Questions**
>
> **Q1 - Intuition behind Theorem 1.** See response to soundness.
>
> **Q2 - Clarification of notation and definition.**  As mentioned in the Response to Presentation, these are defined in the paper. The matrices $\mathbf{H}_k$ are the graph filter coefficients (i.e. the learnable parameters of the GNN). $\mathbf{Z}$ are the input signals to the GNN. If there is data, this input can be the node features $\mathbf{X}$. Alternatively, the inputs can be sampled from a Gaussian distribution, passed through the GNN, and pooled to recover permutation equivariance (RPEARL).
>
> **Q3 - Impact statement**
>
> Done. We added the ICML impact statement to the updated version of the manuscript. We thank the reviewer for pointing this out.

---

> > ### Author Rebuttal · Reviewer_7d7z · 2026-04-01
> >
> > I thank the authors for the clarifications. After reading also other reviews, I am convinced on increasing the score.

---

> > > ### Author Response · Authors · 2026-04-07
> > >
> > > We are glad to hear that we have clarified all of the reviewer's concerns, and are grateful for increasing the score.

---

### Official Review · Reviewer_5Zub · 2026-03-07

**Soundness:** 3
**Presentation:** 3
**Significance:** 3
**Originality:** 3
**Overall Recommendation:** 4
**Confidence:** 4

**Summary:**

This paper proposes a Graph Transformer (GT) framework that incorporates transferable positional encodings (RPEARL and masking) to address training and generalization issues in large-scale graph settings. Through theoretical analysis, the authors show that a GT equipped with transferable positional encodings can inherit the transferability of the encoding itself, enabling a model trained on smaller graphs to achieve competitive performance on larger graphs.

**Compliance With Llm Reviewing Policy:**

Affirmed.

**Final Justification:**

The authors have resolved my concerns. I'd like to keep my positive rating.

**Key Questions For Authors:**

Please refer to the Weaknesses.

**Limitations:**

Please refer to the Weaknesses.

**Strengths And Weaknesses:**

Strengths:

1.	The theoretical analysis is relatively thorough and well-structured, formally demonstrating that graph Transformers can inherit transferability guarantees from their positional encodings under spectral and manifold assumptions.

2.	The experimental evaluation is fairly comprehensive, including comparisons among DenseGT,SparseGT, GNN, Exphormer, and MLP baselines, along with ablation studies that clearly illustrate the contribution of different components, thereby providing empirical support for the proposed approach.


Weaknesses:

1.	The theoretical framework assumes that node features are independently and identically sampled from an underlying manifold, and it remains unclear to what extent this assumption holds for real-world graph data.

2.	Although the introduction claims that GNNs are justified as a reasonable choice for positional encoding, the paper does not provide a dedicated theoretical or empirical comparison demonstrating the necessity or superiority of GNN-based encodings over alternative designs.

3.	In the Sparse GT architecture, the neighborhood hop parameter is evidently a critical hyperparameter that may significantly affect performance, yet the paper lacks a systematic analysis of its sensitivity and its impact on transferability.

4.	There are several instances of unclear logical exposition and minor typographical errors that should be revised for clarity and precision.

---

> ### Author Rebuttal · Authors · 2026-03-31
>
> **Response to weaknesses**
>
> **W1 - Manifold Assumption.** Please see response psEB W2 on the impact of manifold assumptions.
>
> **W2 - Necessity of GNN encodings.** Here we just focus on the transferability of these encodings. See Kanatsoulis for theoretical arguments in favor of GNN as PEs (stable, transferable, permutation equivariant).
>
> We thank the reviewer for this helpful comment. We agree that the current paper does not establish that GNN-based positional encodings are uniquely necessary or uniformly superior to all alternative PE designs. Our theoretical result is in fact more general: the transferability guarantee of graph transformers applies to any positional encoding that satisfies Assumption 3, namely, transferable positional encodings across graph sizes. GNN-based positional encodings are a particularly natural choice in our transferability framework because existing convergence results make their size-transferability analyzable. However, our theory is not restricted to GNN-based encodings and can in principle apply to other PE designs that satisfy Assumption 3. More specifically, RPEARL preserves key properties such as permutation equivariance, expressivity, and transferability. Empirically, our current ablation shows that RPEARL improves over a no-PE baseline and over random-edge variants, but we agree that this is not a comprehensive comparison against other PE families. We will revise the introduction to make this scope explicit and position GNN-based PEs as a principled and analyzable choice within our framework, rather than as the only viable one. Additionally, please refer to the responses PtBR (W1 - Other PEs) for additional discussion and new experiments on alternative PEs (Laplacian PES, RWPEs)
>
> **W3 - K-hop ablation.** While the objective of Sparse GT is to further reduce computational complexity, we agree that the sensitivity neighborhood parameter $K$ is worth exploring. Here we provide an ablation on the arxiv-year dataset, where we observe little sensitivity to $K$ in terms of transferability for values of $K\in\\{1,2,3,4\\}$  [Transferability Plot](https://i.postimg.cc/XYnQ3sKL/ablation-k-arxiv-year.jpg). The optimal value of $K$ may be data dependent, and a further analysis of this and other architectural choices are an interesting future direction of work.
>
> **W4 - Typos and errors.** We have addressed some typos raised by Reviewer 7d7z. We are glad to address any other that the reviewer has identified.

---

> > ### Author Rebuttal · Reviewer_5Zub · 2026-04-01
> >
> > The authors have addressed my questions and concerns. I will keep my positive rating.

---

> > > ### Author Response · Authors · 2026-04-07
> > >
> > > We would like to thank the reviewer for their positive appraisal of our work, and for their productive suggestions.

---

### Official Review · Reviewer_psEB · 2026-03-11

**Soundness:** 3
**Presentation:** 3
**Significance:** 3
**Originality:** 3
**Overall Recommendation:** 4
**Confidence:** 4

**Summary:**

The main concept of this paper pertains to establishing theoretical guarantees showing that graph transformers equipped with GNN-based positional encodings can generalize across graph sizes under a manifold limit model. The paper studies graph transformers (GTs) through a manifold convergence perspective. The authors analyze graph transformers that incorporate GNN-based positional encodings—specifically the RPEARL encoding—and show that if the positional encoding is transferable under graph-to-manifold convergence, then the graph transformer inherits this transferability property. The analysis establishes convergence bounds between a discrete graph transformer and a continuous manifold transformer, implying that models trained on smaller graphs can generalize to larger graphs sampled from the same underlying manifold. Beyond the theoretical results, the paper proposes a practical architecture combining graph transformers with RPEARL positional encodings and evaluates its transferability empirically across multiple datasets. Experiments show that graph transformers trained on smaller graphs can achieve comparable performance when evaluated on larger graphs, supporting the theoretical findings.

**Compliance With Llm Reviewing Policy:**

Affirmed.

**Final Justification:**

The authors have resolved my concerns. I'd like to keep my positive rating.

**Key Questions For Authors:**

1. How sensitive is the transferability result to violations of the manifold sampling assumption in real-world graphs?
2. Would the theoretical results extend to other commonly used positional encodings in graph transformers (e.g., Laplacian eigenvectors or random walk encodings)?
3. How does the proposed approach compare empirically with other representative graph transformer architectures [1-2]?

[1] Ladislav Rampášek, Mikhail Galkin, Vijay Prakash Dwivedi, Anh Tuan Luu, Guy Wolf, Dominique Beaini. Recipe for a General, Powerful, Scalable Graph Transformer. NeurIPS’ 22.
[2] Devin Kreuzer, Dominique Beaini, William L. Hamilton, Vincent Létourneau, Prudencio Tossou. Rethinking Graph Transformers with Spectral Attention. NeurIPS’ 21.

**Limitations:**

The paper does not appear to pose significant direct societal risks, as it primarily studies theoretical properties of graph transformers. However, the discussion of limitations could be expanded. In particular, the theoretical guarantees rely on assumptions such as graph-to-manifold convergence and transferable positional encodings, which may not strictly hold for real-world graphs. It would strengthen the paper if the authors briefly discussed the practical implications of these assumptions and clarified how robust the conclusions are when they are only approximately satisfied.

**Strengths And Weaknesses:**

Strengths:
1. The paper presents a principled theoretical framework connecting graph transformers to manifold neural networks through graph-to-manifold convergence. The analysis provides a key conceptual insight that the transferability of graph transformers is largely determined by their positional encodings, which provides a useful lens for analyzing different graph transformer architectures and highlights the importance of structural encodings.
2. The paper conducts comprehensive experiments demonstrating transferability behavior across different datasets and scales.
3. The proposed framework suggests a valuable training strategy: train graph transformers on smaller graphs and deploy them on larger graphs without retraining, which can significantly reduce computational cost in practice.

Weaknesses:
1. While the paper includes comparisons with several baselines, the experimental section could benefit from broader comparisons with other representative graph transformer models.
2. The theoretical analysis is developed under a graph-to-manifold convergence framework and relies on several assumptions, such as the manifold sampling model, Lipschitz or boundedness conditions on operators, and transferable positional encodings. While these assumptions are standard in theoretical work on graph learning, it would be helpful if the paper could provide more discussion on how these conditions relate to practical graph datasets and the extent to which the results may hold when the assumptions are only approximately satisfied.

---

> ### Author Rebuttal · Authors · 2026-03-31
>
> **Response to weaknesses**
>
> **W1 - Comparisons with other models.** We compare with Exphormer in the Appendix, and have added transferability results for GraphGPS and other positional encodings. See response to PtBR (W1 - Other PEs, W2 - Other Backbones) for more details.
>
> **W2 - Manifold Assumption.** We thank the reviewer for this insightful question. In the revised version, we will expand the discussion of how our assumptions relate to practical graph datasets and how the results should be interpreted when these assumptions hold only approximately. In particular, the manifold sampling model should be viewed as an idealized but meaningful approximation for many real-world settings, including applications such as dynamical systems and images, where the observed graph is often governed by a low-dimensional latent geometry. We believe that this perspective is also reasonable for the real-world graphs used in our node prediction experiments. To support this point empirically, we will add plots of the largest eigenvalues of the graph Laplacian for each dataset. A clear concentration of the spectrum in relatively few modes would be consistent with low intrinsic dimensionality and therefore provide evidence for the manifold viewpoint. We will also clarify that the Lipschitz and boundedness assumptions are regularity conditions used to ensure stability of the limiting operators, while the transferability assumption on positional encodings formalizes consistency across graph sizes. More broadly, although these assumptions may not hold exactly on practical datasets, they are often reasonable approximations, especially with the support of the previous works on stability of the underlying manifold neural networks that provide the guarantee that small fluctuations of the underlying manifold models will not lead to catastrophic performance degradation [1,2]. In such cases our theory could be understood as providing support for approximate, rather than exact, transferability.
>
> [1] Wang, Zhiyang, Luana Ruiz, and Alejandro Ribeiro. "Stability to deformations of manifold filters and manifold neural networks." IEEE Transactions on Signal Processing 72 (2024): 2130-2146.
>
> [2] Wang, Zhiyang, Juan Cerviño, and Alejandro Ribeiro. "Generalization of graph neural networks is robust to model mismatch." Proceedings of the AAAI Conference on Artificial Intelligence. Vol. 39. No. 20. 2025.
>
> **Response to questions**
>
> **Q1 - Sensitivity to manifold assumption.** Please see our response to W2 for further discussion of the impact of the manifold assumptions. With respect to the sampling assumption, our theory assumes that graph nodes are sampled according to the underlying probability distribution on the manifold. In practice, if a large graph already serves as a sufficiently accurate approximation of the latent manifold structure, then uniform subsampling provides a natural proxy for this model. When this assumption is only approximately satisfied in real-world subsampled graphs, we expect the result to remain reasonably robust to mild deviations based on the previous robustness related paper [1,2]. In particular, if the graph operators and positional encodings remain approximately stable across graph sizes, then the graph transformer should still exhibit approximate transferability. We will add a remark to address this impact on the manifold assumption and the sampling sensitivity in the revised version.
>
> **Manifold hypothesis.** Our experiments show that our expected transferability results hold in graphs from different domains (citation graphs, social networks, terrain graphs), which confirms that our assumptions hold in real-world graphs.
>
> **Q2 - Other PEs.** Yes, the theoretical results can be extended to other commonly used positional encodings as long as they satisfy the size transferability Assumption 3 in graph transformers. We have provided theoretical discussion and additional results for eigenvectors and RW PEs. See response to Reviewer PtBR, Q1 for more details.
>
> **Q3 - Other Architectures.** We have added an empirical transferability analysis for GraphGPS. Please see the response to Reviewer PtBR W3. With respect to SAN, its high computational complexity makes it ill-suited for the large scale transferability experiments in our framework. Indeed, note that SAN’s experiments only cover molecule datasets with small graphs. We would also like to clarify that the objective of including different architectures in our experiments is not to compare the performance of graph transformers or propose a new graph transformer architecture, but rather to show the size transferability of graph transformers under our framework.

---

> > ### Author Rebuttal · Reviewer_psEB · 2026-04-01
> >
> > The authors have solved my questions and concerns. I'd like to keep my positive score.

---

> > > ### Author Response · Authors · 2026-04-07
> > >
> > > We thank the reviewer for the productive discussion and for acknowledging that our rebuttal fully addresses their questions.

---

### Official Review · Reviewer_PtBR · 2026-03-13

**Soundness:** 2
**Presentation:** 2
**Significance:** 2
**Originality:** 2
**Overall Recommendation:** 3
**Confidence:** 3

**Summary:**

Graph Transformers (GTs) extend attention-based models to graph-structured data by incorporating GNN-based positional encodings that capture structural information. This work shows theoretically that GTs inherit transferability guarantees from these encodings, allowing models trained on small graphs to generalize to larger ones. Experiments on standard benchmarks and real-world shortest path tasks demonstrate that GTs exhibit scalability comparable to that of GNNs.

**Compliance With Llm Reviewing Policy:**

Affirmed.

**Final Justification:**

The paper establishes a theoretical connection between Graph Transformers (GTs) with GNN-based positional encodings and Manifold Neural Networks (MNNs), which is interesting from a theoretical perspective. However, the practical significance of this contribution appears limited. The notion of transferability has already been extensively studied in the context of GNNs, and the current work mainly extends existing results rather than providing fundamentally new insights.

**Key Questions For Authors:**

The theoretical analysis mainly focuses on the RPEARL positional encoding. Do the proposed theoretical results extend to other commonly used positional encodings in Graph Transformers (e.g., Laplacian PE, random walk PE, or learned structural encodings)?

**Limitations:**

no discussion

**Strengths And Weaknesses:**

Strengths:

This paper establishes a theoretical connection between Graph Transformers (GTs) with GNN-based positional encodings and Manifold Neural Networks (MNNs), providing a useful theoretical perspective for understanding GTs.

Experimental results on multiple large-scale graph datasets demonstrate that GTs equipped with RPEARL positional encodings exhibit promising transferability properties.

Weaknesses:

The study only considers the RPEARL positional encoding, which limits the generality of the conclusions.

The influence of different GT backbones on transferability remains unclear. It would be beneficial to evaluate additional GT architectures in the experiments.

Figure 3 is too small and difficult to read, which affects the clarity of the presentation.

---

> ### Author Rebuttal · Authors · 2026-03-31
>
> **Response to weaknesses**
>
> **W1 - Other PEs.** Our transferability theory is general and applies to any PE that satisfies Assumption 3 (Transferable PE). This covers any GNN-Transformer hybrid, e.g. GraphGPS. Please refer to our answer of Q1for more details on other PEs. We would like to note that our theory is broader than a single encoding: our analysis shows that graph transformers inherit transferability from their positional encodings, while Assumption 3 only requires the positional encoding to be transferable and to converge to a stable limit object across graph sizes. We focus on RPEARL because it is a concrete positional encoding for which Assumption 3 can be justified through existing GNN-to-manifold convergence results, making the end-to-end theorem directly analyzable in our setting. We will therefore clarify that the broader theoretical message extends to other structural positional encodings whenever they satisfy Assumption 3. We explained more in the response to Q1.
>
> **W2 - Other Backbones.** Note we already had Exphormer in the appendix. Here, we provide additional results with GraphGPS. Notice that GraphGPS uses either RW or Laplacian eigenvectors. Furthermore, instead of proposing a better graph transformer architecture, our focus is to show the size transferability of the graph transformer architecture to address the bottleneck of computational efficiency.
>
> **Transferability of Graph GPS**
>
> The following figure shows transferability results for GraphGPS (with Laplacian PEs), compared to dense and sparse GT. GraphGPS exhibits a certain degree of transferability, although we observe less transferability SGT with RPEARL. [Transferability of transformer backbones](https://i.postimg.cc/QNRQNfDc/transformer-architectures.jpg)
>
> **W3 - Figure size.** We will improve the readability of this figure in the updated manuscript. Thank you for pointing this out.
>
> **Response to Questions**
>
> **Q1 Transferability of other PEs.** Our results are general in the following sense: if Assumption 3 holds, namely, if the positional encoding (PE) converges to a limit object and is therefore transferable across graph sizes, then the resulting graph transformer is also transferable. In the case of Laplacian PE, structural information is encoded by the leading eigenvectors of the graph Laplacian, so the PE can be viewed as a linear function of these eigenvectors. Existing convergence results for sampled graphs show that these eigenvectors converge to the corresponding eigenfunctions of the Laplace–Beltrami operator on the underlying manifold. Consequently, Laplacian PE is size-transferable and thus satisfies Assumption 3. It follows that graph transformers equipped with Laplacian PE are also size-transferable. However, as discussed in (Kanatsoulis, 2025), eigenvector-based approaches struggle to be simultaneously expressive, stable under perturbations, and sign-invariant.
>
> Similarly, Random Walk PEs can be shown to be structurally equivalent to  a single layer GNN with no nonlinearity with Random walk transition matrix shift $\mathbf{S} = \mathbf{I-L}$. The aforementioned convergence results may be leveraged to satisfy Assumption 3. However, This positional encoding will be more limited in expressive power than a GNN-based PE.
>
> **Empirical analysis of Laplacian PEs and eigenvector PEs.**
>
> Here we provide experimental results for arxiv-year using Dense GT with alternative positional encodings (Random Walk PE, Laplacian eigenvector PE). The PEs have some degree of transferability, yet they do not transfer to the same rate as RPEARL-based encodings. This result supports our argument that Laplacian and RWPEs transfer but have limited expressivity.
> [Transferability plot of Dense GT with Laplacian PE, Random Walk PE and RPEARL PE](https://i.postimg.cc/HWyYVZ2J/e30TR-posenc-ablation-spectral.jpg).
>
> **RPEARL with different choices of GNN architecture.**
>
> We now present Dense GT with RPEARL PEs using different GNN architectures (GCNConv,SAGEConv, GAT, GIN). We observe that all RPEARL-based PEs exhibit consistent size transferability.
> [Transferability plot of Dense GT backbone with RPEARL PE using different GNN architectures](https://i.postimg.cc/P5z0CcNB/e30TR-posenc-ablation-rpearl-variants.jpg)

---

> > ### Author Rebuttal · Reviewer_PtBR · 2026-04-04
> >
> > I appreciate the authors’ efforts. The response argues that the theory applies to any positional encoding (PE) satisfying Assumption 3; however, it remains unclear how restrictive this assumption is in practice. Additionally, the response suggests that Laplacian and random walk-based PEs are transferable but less expressive. Could the authors provide a more concrete characterization of this trade-off?

---

> > > ### Author Response · Authors · 2026-04-05
> > >
> > > This is a great point. We consider two classes of PE that satisfy Assumption 3: PE based on eigenvectors and PE based on graph neural networks (GNNs).
> > >
> > > PE based on eigenvectors satisfy Assumption 3 because under suitable conditions, eigenvalues and eigenvectors of the graph converge to eigenvalues of the manifold. PE based on GNNs satisfy Assumption 3 for essentially the same reason. Since layers of a GNN are known to be pointwise operators in the graph's spectrum, they satisfy Assumption 3 because eigenvalues (therefore eigenvectors) converge. Nonlinearities transfer because they are pointwise, therefore independent of the graph's structure.
> > >
> > > We advocate for the use of RPEARL because of its **representation** and **stability** properties, but transferability also holds for PE based on eigenvectors or GNNs. In particular, they hold for Laplacian and random walk-based PEs as the reviewer points out.
> > >
> > > Another reason for preferring RPEARL is that RPEARL embeddings are more stable to perturbations, which in this paper implies "better" transferability. I.e., Laplacian and random walk-based PEs satisfy Assumption 3 but with a larger constant. We remark that this is an experimental observation (corroborated by our [experimental results on arxiv-year (plot)](https://urldefense.com/v3/__https://postimg.cc/G9bwfj6r__;!!Mih3wA!Axfo18BtqHBLH6lcYdJXgeAMr7eLaklL0XKzloR6oSzGjvP3lx7C7I8cPcM7F_Ms6F7_Xox0-ixS0EObk_bn5HQ$), RPEARL PE achieves higher accuracy on all training fractions larger than 5%) which is consistent with the instability of eigenvectors to perturbations of a matrix. As it follows from the Davis-Kahan Theorem, eigenvector perturbations can be large even when matrix perturbations are small.
> > >
> > > The better representation or expressivity of RPEARL has been discussed in detail in [1] and we see this as an advantage compared with Laplacian eigenvector and random walk-based PEs. We also observe these advantages of RPEARL from our [comparison with Laplacian and Random Walk PEs (plot)](https://urldefense.com/v3/__https://postimg.cc/G9bwfj6r__;!!Mih3wA!Axfo18BtqHBLH6lcYdJXgeAMr7eLaklL0XKzloR6oSzGjvP3lx7C7I8cPcM7F_Ms6F7_Xox0-ixS0EObk_bn5HQ$): at peak training size (100%), RPEARL PEs achieve higher performance than RWPE or eigenvector PEs.
> > >
> > > We will clarify these points in the final version. We thank the reviewer for the engaging discussion.
> > >
> > >
> > > ### Reference
> > > [1] Kanatsoulis, C., Choi, E., Jegelka, S., Leskovec, J., & Ribeiro, A. (2024, October 4). _Learning Efficient Positional Encodings with Graph Neural Networks_. The Thirteenth International Conference on Learning Representations.

---

### Decision · Program_Chairs · 2026-04-30

**Decision:**

Reject

**Comment:**

Graph Transformers (GTs) extend attention-based models to graph-structured data by incorporating GNN-based positional encodings that capture structural information. This work shows theoretically that GTs inherit transferability guarantees from these encodings, allowing models trained on small graphs to generalize to larger ones. Experiments on standard benchmarks and real-world shortest path tasks demonstrate that GTs exhibit scalability comparable to that of GNNs.  Although it has novel idea and interesting theoritical analysis, it still has some obvious disadvantages as authors pointed out.  It only considers the RPEARL positional encoding, which limits the generality of the conclusions. Moreover, the Impact statement is missing.